# Topological metal-insulator transition within the ferromagnetic state

Ola Kenji Forslund [1,2] ✉, Chin Shen Ong [2] ✉, Moritz M. Hirschmann [3], Nicolas Gauthier [4], Hiroshi Uchiyama [5], Christian Tzschaschel [1], Daniel G. Mazzone [6], Romain Sibille [6], Antonio M. dos Santos [7], Masafumi Horio[1], Elisabetta Nocerino [6,8], Nami Matsubara[9], Deepak John Mukkattukavil[2], Konstantinos Papadopoulos[10], Kazuya Kamazawa[11], Kazuhiko Ikeuchi[11,12], Hidenori Takagi [13], Masahiko Isobe [13], Jun Sugiyama[11], Johan Chang [1], Yasmine Sassa [9], Olle Eriksson [2,14] ✉ & Martin Månsson [9] ✉

A major challenge in condensed matter physics is integrating topological phenomena with correlated electron physics to leverage both types of states for next-generation quantum devices. Metal-insulator transitions are central to bridging these two domains while simultaneously serving as on-off switches for electronic states. Here, we demonstrate how the prototypical material of $K_2Cr_8O_{16}$ undergoes a ferromagnetic metal-insulator transition accompanied by a change in band topology. Through inelastic x-ray and neutron scattering experiments combined with first-principles theoretical calculations, we show that this transition is not driven by a Peierls mechanism, given the lack of phonon softening. Instead, we establish the transition as a topological metal-insulator transition within the ferromagnetic phase with potential axionic properties, where electron correlations play a key role in stabilizing the insulating state. These results reveal how a metal-insulator transition provides a pathway through which magnetism, topology, and electronic correlations interact.

A major milestone in condensed matter physics is bridging the gap between topological phenomena and correlated electron physics to harness both magnetic and topological states for next-generation quantum devices. Central to this effort are metal-insulator transitions (MITs), which serve as both conceptual and functional bridges between these two domains while simultaneously acting as ideal on-off switches for manipulating electronic states, an essential mechanism for device engineering.

While most known MITs occur in antiferromagnetic states with no net magnetization, ferromagnetic MITs, which maintain net

[1]Physik-Institut, Universität Zürich, Zürich, Switzerland. [2]Department of Physics and Astronomy, Uppsala University, Uppsala, Sweden. [3]RIKEN Center for Emergent Matter Science (CEMS), Wako, Japan. [4]Institut National de la Recherche Scientifique - Énergie Matériaux Télécommunications, Varennes, QC, Canada. [5]SPring-8/JASRI, Sayo, Japan. [6]PSI Center for Neutron and Muon Sciences, 5232 Villigen PSI, Switzerland. [7]Neutron Scattering Division, Oak Ridge National Laboratory, Oak Ridge, TN, USA. [8]Department of Chemistry, Stockholm University, Stockholm, Sweden. [9]Department of Applied Physics, KTH Royal Institute of Technology, Stockholm, Sweden. [10]Department of Physics, Chalmers University of Technology, Göteborg, Sweden. [11]Neutron Science and Technology Center, Comprehensive Research Organization for Science and Society (CROSS), Tokai, Japan. [12]Institute of Materials Structure Science, High Energy Accelerator Research Organization, Tsukuba, Ibaraki, Japan. [13]Max Planck Institute for Solid State Research, Stuttgart, Germany. [14]Wallenberg Initiative Materials Science for Sustainability, Uppsala University, Uppsala, Sweden. ✉e-mail: ola.forslund@physics.uu.se; chinshen.ong@physics.uu.se; olle.eriksson@physics.uu.se; condmat@kth.se

magnetization across the transition, offer unique functionality for quantum information and spintronics. However, such transitions are exceptionally rare and poorly understood, particularly when involving topological character. Most well-studied topological phase transitions (TPTs) rely on the closing and reopening of a local gap at a Weyl point, characteristic of single-particle Weyl semimetals[1–3]. In contrast, a more profound form of TPT involves a correlation-driven MIT, where the gap closes without reopening – a phenomenon largely unexplored in topological systems. Such ferromagnetic MITs expand the frontier of topologically driven material manipulation, presenting unprecedented opportunities for innovation in systems where strong electron correlations intersect with topological phenomena[4–10].

In a typical Weyl semimetal, Weyl points exhibit chiral symmetry, meaning points of opposite chirality do not interact[11]. Pairing interactions that break this symmetry, such as charge density waves (CDWs), can induce new phases with topological Goldstone modes known as axions. When $\mathbf{q}_{CDW}$ nests Weyl points of opposite chirality near the Fermi level[11], it can drive a TPT from a Weyl semimetal to a topological axion insulating phase, marking a MIT.

Traditional models, such as the Mott-Hubbard model, have successfully described MITs in various materials[12,13], including high-temperature cuprate superconductors[14] or Verwey transitions in magnetites[15]. However, understanding the temperature-dependent ferromagnetic-metal-to-ferromagnetic-insulator transition (FM-MIT) has proven particularly challenging. $K_2Cr_8O_{16}$ is renowned for exhibiting this FM-MIT. It has a Curie temperature $T_C = 167$ K with a MIT ($T_{MIT} = 95$ K) that retains the FM order[16]. Since $K_2Cr_8O_{16}$ is considered a quasi-1D compound (Fig. 1), the MIT has been explained by a 1D CDW[17], also known as a Peierls transition, accompanied by a structural distortion from a metallic tetragonal phase to an insulating monoclinic phase (Fig. 1) with $\mathbf{q}_{CDW} = (1/2, 1/2, 0)$[17,18]. However, a recent study[19] suggesting the presence of Weyl fermions raises new questions regarding the role of topological effects in this transition. Related compounds, such as $RbCr_4O_8$, have also been shown to host Weyl points[20], demonstrating the robustness of these topological features in this family of compounds.

Here, we show that the ferromagnetic MIT in $K_2Cr_8O_{16}$ is associated with a change in band topology. Using a combination of neutron diffraction, inelastic x-ray scattering, and first-principles calculations, we demonstrate that no phonon condensation is observed across the transition. Instead, Weyl fermions of opposite chirality are nested by the CDW wavevector. These results are inconsistent with a Peierls driven mechanism and instead identify the transition as a correlation-driven MIT occurring within the ferromagnetic phase, with possible axionic character.

## Results

### Magnetic response across the transition

As the positions of the Weyl points are highly dependent on the FM ordering[19], we investigated the magnetic response across the phase transitions using powder and single-crystal neutron scattering. The collected ND data and the corresponding Rietveld refinement at $T = 10$ K are shown in Fig. 2a. Upon cooling below $T_C = 167$ K, a strong enhancement of the {1,2,1} reflections is observed, consistent with additional scattering resulting from long-range FM ordering with $\mathbf{q}_{FM} = (0, 0, 0)$ where $\mathbf{q}_{FM}$ is the magnetic propagation vector. In order to identify the magnetic structure, the group-subgroup relationships for the given magnetic propagation were analyzed (Sec. S2 of Supplementary Materials (SM)), and the solution of $C2'/m'$ (#12.62) was obtained (Fig. 1(b)). The FM spin polarization reduces the symmetry from the paramagnetic space group $I4/m$ (#87) to the magnetic subgroup $C2'/m'$ (#12.62), which is monoclinic. The moment direction within the $ab$-plane could not be determined due to the powder average, an ambiguity that could not be resolved even with single-crystal ND due to the formation of magnetic domains. The magnetic {1,2,1} peaks could in principle be inequivalent below the temperature of the MIT ($T_{MIT}$), due to the monoclinic distortion. However, temperature-dependent measurements of the peaks (1,2,1) and (2,1,1) do not show a difference (Fig. 2b). To resolve the magnetic order within the $ab$-plane, angle dependent magnetization measurements in the $ab$ plane as a function of magnetic field at $T = 120$ K is presented in Fig. 1c. Fitting the data to a sinusoidal function yields an order direction tilted ~22.5° relative to the principal axis. Together, these results show that the magnetic order remains unchanged across the MIT, while the moment orientation at 120 K lies slightly off-axis within the $ab$-plane.

To further study the spin correlations, inelastic neutron scattering (INS) spectra were measured on powder at $T = 5$ and 130 K (Fig. 2c, d). Both spectra contained the same essential features, confirming the absence of any significant change in the exchange interactions across the MIT. Figure 1b defines the most relevant exchange parameters for the compound: $J_1$ and $J_2$ (between nearest neighbor separated by ~2.9

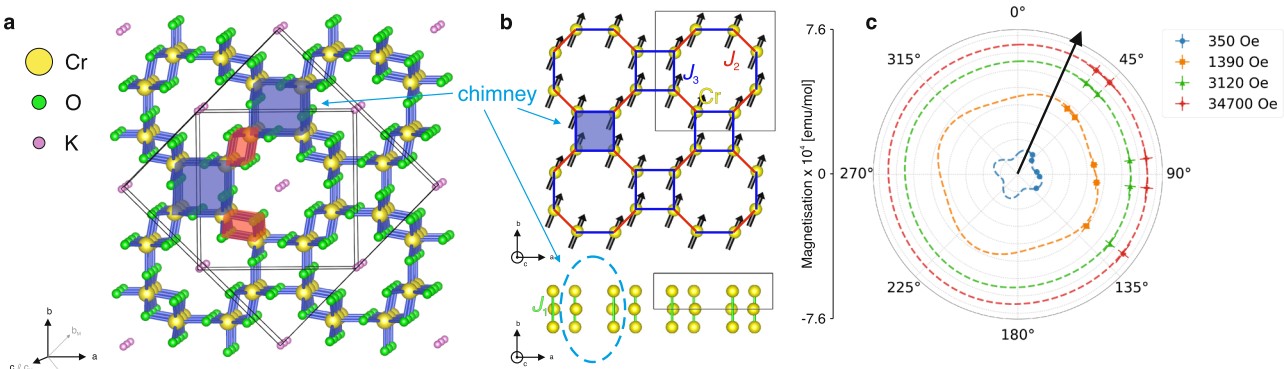

**Fig. 1 | Crystal structure of $K_2Cr_8O_{16}$. a** Tetragonal (smaller) and monoclinic (larger) crystal structures of $K_2Cr_8O_{16}$, outlined by solid boundaries. The chimney building blocks consist of a square arrangement of four corner-sharing $CrO_6$ octahedra that propagate along the $c$ axis. The lattice vectors (**a**) (**a**$_M$), (**b**) (**b**$_M$), and (**c**) (**c**$_M$) denote the tetragonal (monoclinic) phase, respectively. Double-chain (inter-chimney) interactions arise from Cr-Cr bonding through edge-sharing $CrO_6$ octahedra. **b** Structural model showing only the magnetic Cr atoms. The bonding interactions $J_1, J_2,$ and $J_3$ are indicated, where $J_1$ and $J_3$ correspond to intra-chimney interactions and $J_2$ corresponds to the inter-chimney interaction. The magnetic structure determined from neutron diffraction and angle-dependent magnetization measurements is shown by arrows. **c** Angle-dependent magnetization measured at $T = 120$ K, plotted in units of emu/mol for several applied magnetic fields. Symbols denote data collected at different applied magnetic fields (see legend). Dashed lines represent sinusoidal fits to the data. The arrow marks the easy-axis orientation, approximately 22.5° relative to the principal axis (0°).

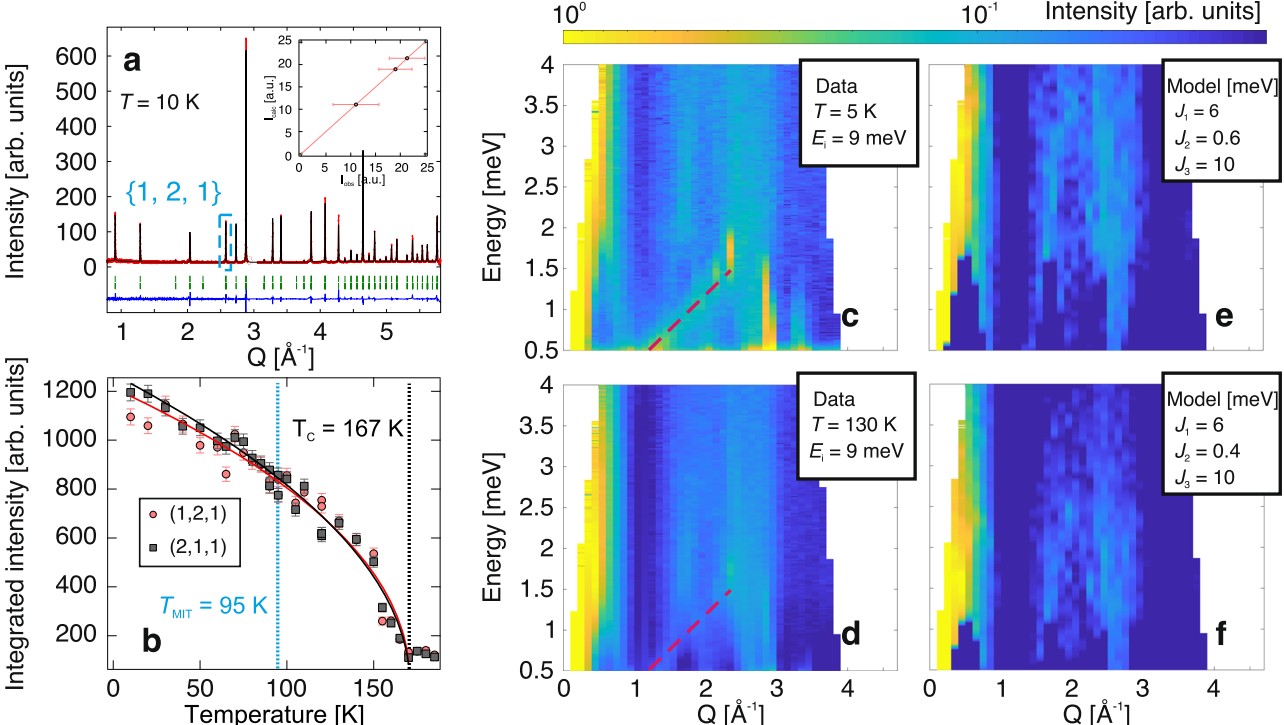

**Fig. 2 | Neutron scattering data of $K_2Cr_8O_{16}$. a** Neutron powder diffraction pattern collected at $T = 10$ K. Experimental data are shown in red, while the calculated refinement profile is shown in black. Tick marks indicate the allowed reflections, and the lower trace shows the difference between the calculated model and the experimental data. The main magnetic peak {1, 2, 1} is indicated (see Supplementary Fig. S3 for high-temperature patterns). The inset compares the measured ($I_{obs}$) and calculated ($I_{calc}$) intensities from single-crystal magnetic refinement using datasets obtained by subtracting selected peaks at 10 K from those at 200 K. **b** Temperature dependence of the neutron diffraction peak intensities at the (2, 1, 1) (red circles)

and (1, 2, 1) (black squares) reflections. $T_{MIT}$ denotes the metal-insulator transition temperature. Solid lines represent fits using $I = I_0(1 - T/T_C)^\beta$. **c,d** Powder inelastic neutron scattering spectra collected with an incident energy $E_i = 9$ meV at 5 K and 130 K, respectively. Dashed lines indicate an additional dispersion discussed in the Supplementary Information. **e,f** Fits to the inelastic neutron scattering spectra obtained using a Heisenberg Hamiltonian at 5 K and 130 K. **c–f** use the same color scale. The fitting and analysis procedures are provided in the Supplementary materials. Error bars represent the experimental uncertainty.

Å) are both intra-chain interactions within the double chain of edge-sharing octahedra (Fig. 1(a), shaded red), whereas $J_3$ (between third-nearest neighbors separated by ~ 3.4 Å) is the inter-chain interaction between corner-sharing octahedra (Fig. 1a, shaded blue). In this paper, we use the following conventions for the classical Heisenberg Hamiltonian, $\mathcal{H}$, defined as $\mathcal{H} = -\frac{1}{2}\sum_{ij}J_{ij}(\mathbf{e_i} \cdot \mathbf{e_j})$, where $\mathbf{e_i}$ is the unit vector in the direction of the $i$-th site magnetization and $J_{ij}$ is the isotropic exchange interaction parameter multiplied with the square of the magnetic moment of the $i$-th site. With this convention, we note that positive exchange values ($J$) favor FM alignment.[21]

Using linear spin wave theory[22] and the magnetic order derived from ND, the exchange parameters, $J_1 = 6.0$, $J_2 = 0.6$ and $J_3 = 10$ meV are obtained by fitting the experimental INS spectra to the Heisenberg Hamiltonian (Fig. 2e, f and Sec. S2 of SM), consistent with FM spin alignment. The data also confirm that the MIT has little or no effect on the exchange couplings. To verify the experimentally obtained values, we used density functional theory (DFT) and the magnetic force theorem[23,24] to calculate $J$ from first principles. The theoretically calculated exchange parameters are also smaller between the nearest neighbors, $J_1$ and $J_2$, than they are for the third exchange parameter, $J_3$ (Fig. S10(a) SM), in agreement with our experimental data and previous calculations[17]. These results confirm that the magnetic building blocks of the quasi-1D structure are the groups of four corner sharing chains between next-nearest neighbors (referred to as chimneys[17]; Fig. 1(a),(b) in shaded blue), instead of a one-dimensional linear chain (Fig. 1(a),(b)). This questions the previous conclusions that $K_2Cr_8O_{16}$ experiences Peierls instability at MIT, a notion predicated on $K_2Cr_8O_{16}$ being 1D[17] and is further

weakened by the absence of experimental evidence. We will return to this point later in this paper.

Finally, we found a clear $t^2/U$ dependence in the calculated exchange interactions, where $t$ is the strength of electron hopping between sites and $U$ is the Hubbard repulsion of electrons at the same site. This quadratic scaling indicates that the average interactions in $K_2Cr_8O_{16}$ arise from a superexchange mechanism[25,26], rather than double exchange[17,18,27–29] – for which $J_{DE} \propto t$ – or direct exchange type[30]. While superexchange is often associated with antiferromagnetism, in this case the dominant couplings are ferromagnetic, consistent with the Goodenough-Kanamori-Anderson rules. The microscopic origin is outlined in Sec. S3C of the SM.

**Topological phase transition**

Using structural[17,31] and magnetic information derived from experiments, we analyzed the electronic properties by performing DFT calculations and employing a group theory analysis. We start our discussion with the paramagnetic tetragonal metallic phase, with space group of $I4/m$ (#87). In terms of atomic arrangement and structural periodicity, all Cr are symmetrically equivalent. The mirror symmetry relates their $d_{xz}$ and $d_{yz}$ orbitals to each other (Fig. 3a), i.e., in the paramagnetic phase all bands have an equal amount of $d_{xz}$ and $d_{yz}$ characters (Fig. 3(c,d)) (as pointed out in Ref. 27). In the present work, the spatial orientation of the $d$-orbitals are defined with respect to the local $z$-axis of the Cr-O octahedron (as defined in Fig. 3(a)), which due to the crystal symmetry, always lies parallel to the $ab$-plane. The symmetry-equivalent $d_{xz}$ and $d_{yz}$ orbitals are higher in energy than the $d_{xy}$ orbital due to compression along the $z$-axis of the distorted Cr-O

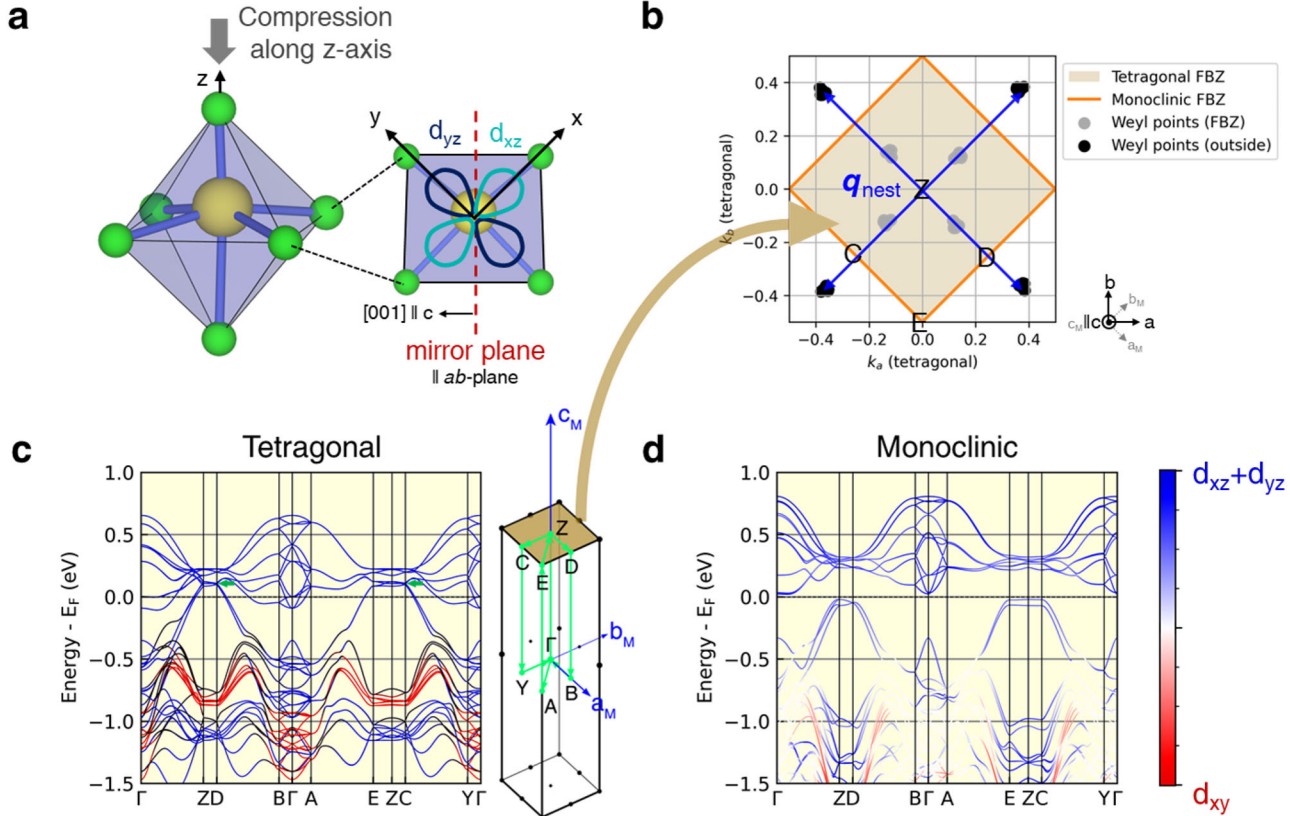

**Fig. 3 | First-principles theoretical calculations. a** One of the distorted Cr-O octahedra, compressed along the $z$ direction, with mirror symmetry parallel to the $ab$ plane. The $x$, $y$, and $z$ axes refer to the local axes of the octahedron. **b** Symbols denote pairs of Weyl points in the first Brillouin zone (FBZ) of the tetragonal phase, calculated using the monoclinic lattice supercell defined by the $\mathbf{a}_M$, $\mathbf{b}_M$, and $\mathbf{c}_M$ lattice vectors. The Weyl points are located on the nodal plane defined by $k_c = \pi/c$. The vector $\mathbf{q}_{nest}$ indicates the nesting between Weyl points. **c,d** Density functional theory band structures of the tetragonal ($U = 0.0$ eV) and monoclinic ($U = 4.0$ eV) phases, respectively, plotted relative to the Fermi level ($E_F$). The color scale indicates the projection onto $t_{2g}$-like Wannier orbitals. Panels (**c**) and (**d**) use the same color scale, shown in (**d**). In (**c**), an arrow marks the approximate location of one of the Weyl points. The inset shows the FBZ of the monoclinic lattice and the **k** path along which the band structures in (**c,d**) are plotted, with the shaded surface indicating the $k_c = \pi/c$ plane shown in (**b**).

octahedra (Fig. 3(a,c,d)). Since Cr in $K_2Cr_8O_{16}$ has a nominal charge of + 3.75, only (6 − 3.75 = ) 2.25 of its $t_{2g}$-like (i.e., $d_{xy}$, $d_{yz}$ and $d_{xz}$) orbitals are occupied. The fractional occupation of these states guarantees that the tetragonal phase is metallic or at least filling-enforced semi-metallic (Fig. 3(c)).

In order to evaluate the topological nature of the bands, we used the DFT wavefunctions to construct a downfolded tight-binding model with the Cr $d_{xy}$, $d_{yz}$ and $d_{xz}$ Wannier functions as basis states. By calculating the Berry curvature of the magnetic metallic phase (#12.62), we found pairs of Weyl points located close to $k_c = \pm \pi/c$ nodal plane 0.1 eV above the Fermi level (see Fig. 3(b,c) and see Table S3 SM for their precise locations)), confirming the topological nature of the electronic structure. For each pair, Weyl points of opposite chiralities are related by inversion symmetry, consistent with the magnetic space group #12.62. Furthermore, since all irreducible co-representations of this group are one-dimensional, the Weyl points are accidental crossings near the $k_c = \pm \pi/c$ plane, which explains why the Weyl points are not located at high-symmetry **k**-points in the BZ (see Table S3 SM for the locations). In addition, we find that the positions of the nodal points are robust. Regardless of the direction of the magnetic moment, we found that the nodal points always form a cross-like feature about the nodal plane, such that nested nodal pairs are always present in the (0.5, 0.5, 0.0) and ( − 0.5, − 0.5, 0.0) directions of the tetragonal Brillouin zone (SM Fig. S8).

These Weyl points exhibit remarkable characteristics. In particular, the nesting vectors connecting Weyl points of opposite chirality, $\mathbf{q}_{nest}$ = (0.75, 0.75, 0.0) and ( − 0.75, − 0.75, 0.0) (Fig. 3(b)), closely

match the lattice distortion wavevector observed in single-crystal X-ray diffraction experiments[17,18]. This suggests that the nesting of Weyl points may be coupled to the lattice distortion, thereby breaking chiral symmetry and allowing the emergence of axion-like excitations[32–34]. A definitive confirmation of this scenario remains an important direction for future investigation.

Secondly, the Weyl points lie near the nodal planes at the Brillouin-zone boundary, $k_c = \pm \pi/c$, where every Bloch state is symmetry-enforced to be doubly degenerate. This degeneracy arises from the magnetic screw rotation $\mathcal{T}C_2^{[001]}$, which combines a 180° rotation about the [001] axis with time-reversal symmetry. The operation squares to a lattice translation along **c** with a phase factor, $(\mathcal{T}C_2^{[001]})^2 = e^{ik_c c}$, so at $k_c = \pm \pi/c$, it imposes $(\mathcal{T}C_2^{[001]})^2 = -1$, enforcing degeneracy. This protection is a direct consequence of the base-centering in the conventional unit cell, which introduces a fractional translation in the symmetry operation. As a result, the $k_c = \pm \pi/c$ planes host symmetry-enforced nodal degeneracies that constrain the electronic band structure and promote the formation of Weyl points.

Having discussed the metallic phase, we now proceed to analyze the insulating phase, which belongs to the structural space group $P2_1/c$ (#14). In this phase, the degeneracy between the $d_{xz}$ and $d_{yz}$ orbitals is lifted due to monoclinic distortions. Moreover, all four Cr atoms within each $Cr_4O_6$ chimney become inequivalent, further splitting the $d$-orbital levels into twelve non-degenerate states. As a result, the $d$ electrons can redistribute unevenly among the Cr sites. A nominal filling of 2.25 electrons per Cr corresponds to nine out of 12 available $t_{2g}$-like states being occupied. This is consistent with the calculated (2.1

$\mu_B$) and experimentally measured (2.2 $\mu_B$) Cr magnetic moments. Moreover, the total electron count and the total spin moment per primitive cell are both integers, a condition necessary for the opening of a band gap.

Since the magnetic moment directions are confined to the $ab$-plane, we deduced the highest symmetry magnetic subgroup to be $P2_1'/c'$ (#14.79). This subgroup is different from metallic phase magnetic group $C2'/m'$ (#12.62) in one distinct way: the monoclinic subgroup of the insulating phase is non-symmorphic, and its primitive unit cell coincides with the conventional one. Consequently, the screw rotation's fractional translation does not arise from lattice centering, and the symmetry-enforced degeneracy on the $k_c = \pm \pi/c$ planes originates directly from the non-symmorphic symmetry, not from band folding.

This reduction of translation symmetry is accompanied by a change in topological character. In the metallic phase, accidental crossings of the Cr $t_{2g}$-derived bands, allowed by the magnetic space group (#12.62), produce Weyl points near the $k_c = \pm \pi/c$ planes. In the insulating phase, the Weyl points are removed due to the combination of monoclinic crystal distortion and orbital splitting. We further note that the nodal planes are topologically uncharged: they are symmetry-enforced by $\mathcal{T}C_2^{[001]}$, and inversion symmetry renders their Chern number trivial. By calculating the parity eigenvalues at all time-reversal invariant momentum points[35], we confirm that the insulating monoclinic phase is a trivial insulator (i.e., $\mathbb{Z}_4$ index = 0). In summary, the disappearance of Weyl points is a direct consequence of translation symmetry reduction due to the tetragonal-to-monoclinic structural distortion, establishing the MIT as a topological phase transition within the FM state.

### Phonon soft mode analysis

We next focus on the driving mechanism behind the FM-MIT. The Peierls instability proposition is based on the experimental observation of superstructure peaks at $\mathbf{q}_{CDW}$ = (0.5, 0.5, 0) below $T_{MIT}$[17,18]. We confirm the existence of these by plotting the temperature dependence of the integrated elastic peak intensity at $\boldsymbol{q} = h\bar{h}0$ for $h$ = 4.50(5) in Fig. 4b. In a Peierls transition[36], the phonon energy of a strictly 1D material softens via the electron-phonon coupling due to divergence in the electronic susceptibility. This divergence causes the phonon energy to renormalise and collapse onto the elastic line (also known as a Kohn anomaly or phonon condensation), forming new Bragg peaks.

In order to determine the role of phonons in a FM-MIT, we present IXS spectra collected at three different temperatures (115, 95 and 20 K): above, around and below the $T_{MIT}$, for a wavevector close to $\mathbf{q}_{CDW}$ ($\boldsymbol{q} = h\bar{h}2$ where $h$ = 4.45) in Fig. 4a. Apart from the elastic peak, the energy scans contain inelastic Stokes and the smaller anti-Stokes pairs, attributed to a phonon mode. The observed phonon mode around $\mathbf{q}_{CDW}$ reveals no temperature dependence ( ~ 10 meV for all temperatures close to $\mathbf{q}_{CDW}$, Sec. S1). This demonstrates the absence of phonon softening at $\mathbf{q}_{CDW}$ during the MIT, firmly establishing that the phase transition is not a Peierls transition.

The absence of phonon softening is more clearly shown in Fig. 4c, d, where the phonon dispersion of $\boldsymbol{q}$-vectors across $\mathbf{q}_{CDW}$ is shown over a wide range. The transverse acoustic phonon branch dispersed linearly with $q$ from the $\Gamma$-point for all temperatures. A similar linear dispersion was observed for the longitudinal acoustic phonon, albeit at a larger gradient with respect to $\boldsymbol{q}$. Nonetheless, irrespective of phonon polarization, we were unable to detect any phonon condensation at $\mathbf{q}_{CDW}$. This is further corroborated by our first principles phonon calculations. The calculated phonon dispersion for the tetragonal structure from $\Gamma$ to M is superposed with experimental data points obtained from IXS with good agreement (Fig. 4e). These results are indirectly supported by a high-pressure experiment, which shows an anti-ferromagnetic ordering stabilized under pressure[37], instead of a FM-like ordering as theoretically predicted[29] for a Peierls insulating FM compound.

We confirm that there is no drastic phonon softening from a higher-energy optical phonons through our Density-Functional Perturbation Theory (DFPT) calculation. Instead, phonon modes in the insulating monoclinic phase exhibit some minor softening for the low-energy Cr phonons relative to the metallic phase (Fig. S11e), leading to higher vibrational entropy. At low temperatures, the insulating phase is favored over the metallic phase as the former will have a lower Gibbs free energy. The fully calculated phonon spectra containing all phonon branches are included in Fig. S11a, b).

Finally, a hard x-ray photoelectron spectroscopy (HAXPES) measurements[38] have showed that in the metallic phase, the valence of Cr is not static at the nominal valence value of 3.75. Instead, a dynamic noninteger $Cr^{4+}$-$Cr^{3+}$ valence fluctuations was observed. In fact, similar valence fluctuations have indeed mediated phase transitions, such as structural instabilities in YbPb, where lattice-valence fluctuations coupling allowed a structural instability[39]. One may speculate that there exists a valence-lattice coupling in $K_2Cr_8O_{16}$ as well, from which the low temperature structure is derived.

## Discussion

The driving mechanism behind the temperature-dependent ferromagnetic MIT in $K_2Cr_8O_{16}$ has been debated since its initial reports. Owing to the quasi-one-dimensional nature of this material, theoretical descriptions have often invoked a Peierls instability to explain the transition. Such a mechanism relies on phonon condensation, which is not supported by the experimental observations presented here. Instead, by combining inelastic scattering experiments with first-principles calculations, we show that the MIT is associated with a change in band topology within the ferromagnetic phase.

Across the transition, a structural distortion of the octahedral environment lifts the degeneracy between the $d_{xz}$ and $d_{yz}$ orbitals, leading to the opening of an energy gap. This same distortion also removes the topological character of the electronic bands, resulting in a MIT that coincides with a topological phase transition in a magnetically ordered state. These results bridge the realms of electron correlations and topological physics, and suggest opportunities for future research on quantum devices that leverage both magnetic and topological states.

## Methods

### Sample synthesis

The difficulty in determining the underlying physics of $K_2Cr_8O_{16}$ is partly related to the small size of the available single crystals, which arises from the required high-pressure synthesis conditions. To address this, new cell designs for a high-pressure Walker-type multi-anvil system were developed at the Max Planck Institute for Solid State Research, enabling growth of larger, higher-quality crystals. The new 18/11 and 25/15 multianvil cell assemblies provide more uniform pressure and temperature distributions, resulting in increased sample volumes (up to several mm in size) while maintaining stable synthesis conditions up to ~ 12 GPa and 1500 °C. With these improved cells the $CrO_2$ impurity phase is reduced to below 2% in powder, and the obtained crystals are significantly larger (up to 0.5 × 0.5 × 1 mm³) with $T_{MIT} \simeq 100\,K$. Although, we have referenced the value of $T_{MIT}$ = 95 K in this work, based on previously reported values.

For this work, two single crystals (0.5 × 0.5 × 1 mm³ and 0.1 × 0.1 × 0.2 mm³) with a $T_{MIT} \simeq 100\,K$ were successfully prepared. These crystals were grown using the high-pressure Walker-type multianvil apparatus. A stoichiometric mixture of $K_2Cr_2O_7$:$Cr_2O_3$ = 1: 3 was loaded into multianvil cell assemblies and compressed to pressures of approximately 7 GPa. The mixture was heated to temperatures up to 1273 K, held at the synthesis temperature for 1 h, and then cooled to room temperature before decompression to ambient pressure. Single crystals were recovered from the reaction products and selected for the experiments.

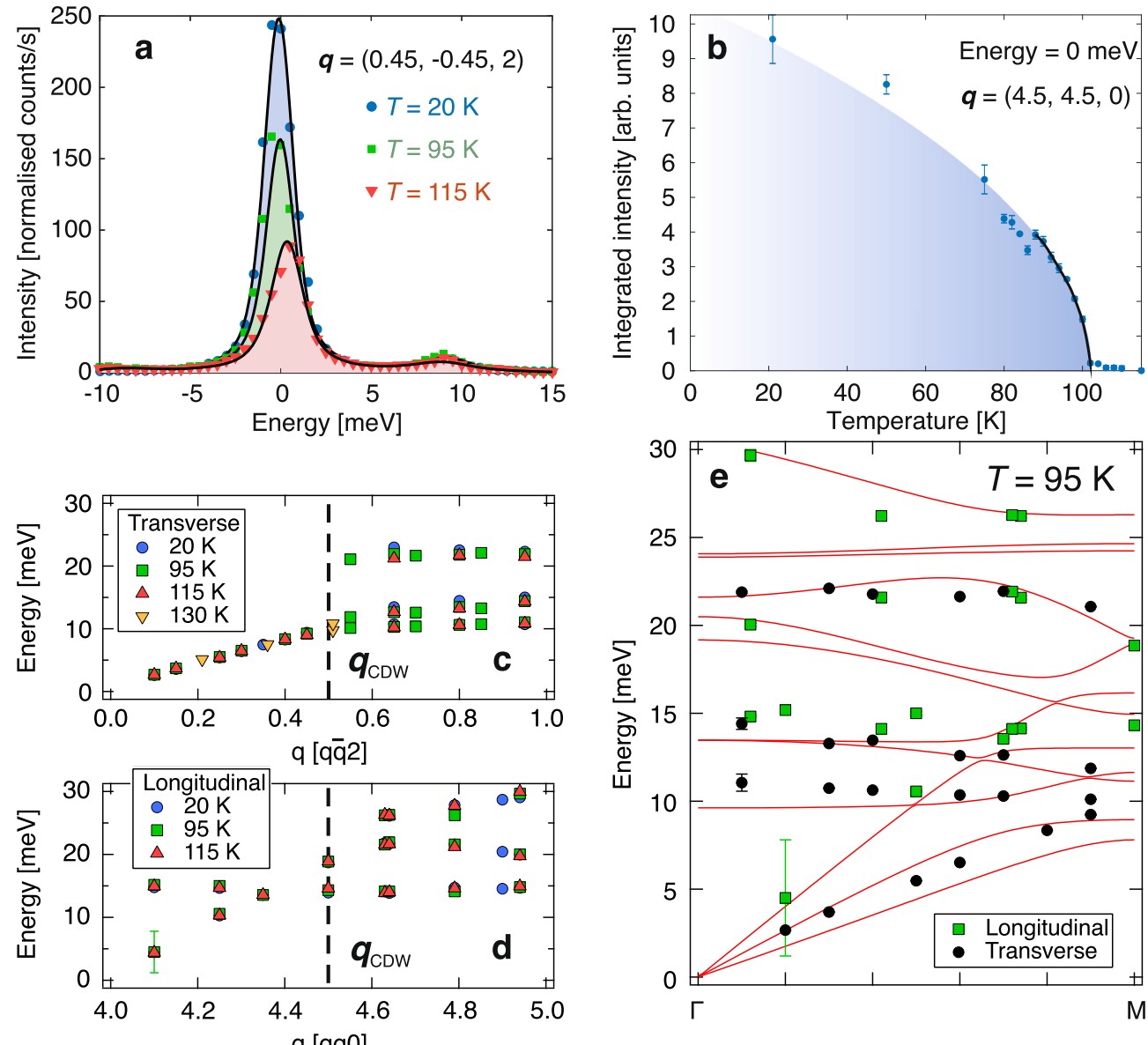

**Fig. 4 | Inelastic x-ray scattering data of $K_2Cr_8O_{16}$. a** Inelastic x-ray scattering spectra collected in transverse geometry at 20 K (circles), 95 K (squares), and 115 K (triangles). The scans were performed from $E = -10$ to 30 meV at $\mathbf{q} = h\bar{h}2$ for $h = 0.45$. Solid lines represent fits to the data (Section 1 in the Supplementary materials). **b** Integrated elastic scattering intensity collected along $\mathbf{q} = h\bar{h}0$ between $h = 4.45$ and 4.55, i.e., across $\mathbf{q}_{CDW}$. Four scans were performed and error bars represent the standard deviation. The solid line is a fit using the mean-field expression $I(T) = I_0(1 - T/T_C)^\beta$. **c,d** Phonon dispersion along $q\bar{q}l$, collected at $T = 20$, 95, and 115 K in transverse and longitudinal geometries, respectively. One scan performed at 130 K is included. Symbols denote data collected at different temperatures (see legend). **e** Experimentally measured phonon dispersion (symbols) and calculated phonon dispersion (solid lines) from $\Gamma$ to M in the folded configuration. Different symbols denote longitudinal and transverse phonon modes. Error bars represent the experimental uncertainty unless otherwise stated.

## Inelastic X-ray scattering measurements

The crystals were mounted onto separate copper sample holders and aligned to measure the transverse and longitudinal phonons using IXS. The IXS data were collected at the high resolution beamline BL35XU at the SPring-8 synchrotron source in Japan using $E_i = 21.74$ keV[40]. The inelastic peaks were fitted using a damped harmonic oscillator convoluted with the experimental resolution ($\sim 1.7$ meV). The peaks were fitted using the software Dave[41].

## Magnetization measurements

Angle-dependent magnetization measurements were performed on a single crystal of $K_2Cr_8O_{16}$ (mass $m = 0.00148$ g) using a Quantum Design MPMS3 magnetometer in DC mode. Measurements were taken at selected angles between 0° and 135° (37°, 45°,

83°, 96°, and 127°) relative to the crystallographic $a$- and $b$-axes at $T = 120$ K under zero-field-cooled conditions. The initial crystal orientation was determined by Laue diffraction, after which controlled relative rotations were applied. The magnet was demagnetized between each angular measurement to minimize remanent fields. Magnetic fields up to 3.5 T were applied, and the measured magnetic moments were converted to molar magnetization. Assuming fourfold rotational symmetry, the angular dependence of the magnetization was fitted with the sinusoidal function; $M(\theta) = A\sin(4\theta + \phi) + M_0$.

## Neutron scattering measurements

Neutron powder diffraction (NPD) was measured at the Spallation Neutron Source (SNS) in Oak Ridge National Lab (ORNL) using the

POWGEN instrument[42], set at a central wavelength of 0.8 Å and a bandwidth of 1 Å. These measurements were made in high resolution mode. About 500 g of sample was loaded under helium gas for temperature equilibration onto a vanadium container. Single-crystal neutron diffraction (ND) was performed at the Paul Scherrer Institut (PSI) using the ZEBRA instrument with wavelength = 1.178 Å on a single crystal (0.5 x 0.5 x 1 mm³). A 4 circle goniometer with a closed cycled refrigerator was used in order to measure selected peaks from 10 K up to room temperature. The inelastic neutron scattering (INS) was measured at Japan Proton Accelerator Research Complex (J-PARC) using the 4SEASON instrument on a powder sample of ~ 2 g with a resolution of 5% of $E_i$[43–45]. Given the uniqueness of the instrument, several $E_i$s (7, 9, 12, 18, 27, 47, 97 and 300 meV) were measured simultaneously at 10, 130 and 200 K and the relevant data is presented in this work. All measured neutron scattering intensities were normalized to the spallation source proton current. The diffraction pattern was analyzed using the Fullprof software package[46] while the inelastic spectra were analyzed using SpinW[47].

### DFT calculations

The DFT calculations were performed using the `Quantum Espresso`[48] package. All DFT calculations for band structures, $J_{ij}$, phonon, Berry curvature and Weyl point analyses were performed in the ferromagnetic spin-polarized state. DFT calculations were carried out with $U = 4.0$ eV for the insulating monoclinic phase to capture electron-correlation effects, whereas $U = 0.0$ eV was used for the metallic phase, where metallic screening is expected to strongly reduce the effective on-site interactions. The phonon eigenvectors and eigenvalues were calculated using the finite displacement and supercell approach with `Phonopy`[49]. Wannier functions are constructed from the DFT eigenvectors and eigenvalues using `Wannier90`[50]. The Cr $t_{2g}$ states were used as initial projections in the local coordinates of the octahedral crystal field. Using these Wannier functions, we use the magnetic force theorem[23,24] to map the tight-binding Wannier Hamiltonian onto a classical Heisenberg model using `TB2J`[51]. The chiralities of the Weyl points were calculated using `WannierTools`[52] with the same Wannier basis sets. For more technical details, see Sec. S3 of SM.

## Data availability

All data needed to evaluate the conclusions of this study are available within the Article and its Supplementary Information. All datasets generated and/or analyzed during the current study are available from the corresponding authors upon request.

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

## Acknowledgements

The IXS measurements were performed at the SPring-8 with the approval of the JASRI (Proposal No. 2019A1525 and 2020A1653). One of the neutron scattering experiments was performed at the MLF, J-PARC, Japan, under a user program (No. 2019A0287). A portion of this research is based on experiments performed at the Swiss spallation neutron source SINQ, Paul Scherrer Institute, Villigen, Switzerland. A portion of this research used resources at the Spallation Neutron Source, a DOE Office of Science User Facility operated by the Oak Ridge National Laboratory. The beam time was allocated to POWGEN on proposal number IPTS-21093.1. This research is funded by the Swedish Foundation for Strategic Research (SSF) within the Swedish National Graduate School in Neutron Scattering (SwedNess), as well as the Swedish Research Council VR (Dnr. 2021-06157 and Dnr. 2022-03936), and the Carl Tryggers Foundation for Scientific Research (CTS-22:2374). O.K.F is supported by the Swedish Research Council (VR) through Grant 2022-06217, the Foundation Blanceflor fellow scholarships for 2023 and 2024, and the Ruth and Nils-Erik Stenbäck Foundation. Y.S. acknowledges support from the Swedish ResearchCouncil (Dnr 2025-08127), and The Knut and Alice Wallenberg Foundation (2021.0150). The theory calculations were supported by the ERC synergy grant (FASTCORR, project 854843) (C.S.O. and O.E.), the Swedish Research Council (C.S.O. and O.E.), WISE, Wallenberg Initiative Materials Science for Sustainability, funded by the Knut and Alice Wallenberg foundation (O.E.) and Swedish National Infrastructure for computing (C.S.O. and O.E.). M.M.H. was funded by the Deutsche Forschungsgemeinschaft (DFG, German Research Foundation) - project number 518238332 and by RIKEN Special Postdoctoral Researcher Program. E.N. acknowledges financial support from the SSF-Swedness grant SNP21-0004 and the Foundation Blanceflor 2024 fellow scholarship. J.S. acknowledges support from Japan Society for the Promotion Science (JSPS) KAKENHI Grants No. JP23K26533 and No. JP24H00042. O.E. acknowledges support from the Swedish Research Council, The Knut and Alice Wallenberg Foundation, WISE - The Wallenberg Foundation Materials Science, eSSENCE, StandUp, and the ERC (synergy grant FASTCORR, project 854843). Calculations performed from resources provided by NAISS. All crystal structure figures were made with VESTA software[53].

## Author contributions

The project was conceived by O.K.F, J.S. and M.M. Sample synthesis was carried out by M.I. and H.T. Inelastic x-ray scattering experiments were performed by O.K.F., H.U., D.G.M., M.H., E.N., N.M., D.J.M., K.P., J.C., Y.S., and M.M. Neutron scattering experiments were conducted by O.K.F., N.G., R.S., A.M.S., E.N., N.M., K.K., K.I., J.S., and M.M. Magnetization measurements were performed by O.K.F. and C.T. The theoretical calculations were performed by C.S.O., M.M.H., O.E. O.K.F., and C.S.O. prepared the manuscript, and all co-authors contributed to the final draft.

## Funding

## Competing interests

The authors declare that they have no competing financial or non-financial interests.

## Additional information

**Peer review information** : *Nature Communications* thanks Norbert Nemes, Alexander Tsirlin, and the other, anonymous, reviewer(s) for their contribution to the peer review of this work. A peer review file is available.

