## [Transparent Peer Review file · Nature Communications]

Topological Metal-Insulator Transition within the Ferromagnetic state

Corresponding Author: Dr Ola Forslund

Version 0:

Reviewer comments:

Reviewer #1

(Remarks to the Author)

This manuscript is a combined computational and experimental work outlining a topological phase transition that occurs beneath (in temperature) a magnetic phase transition to a magnetic state. The compound, $K_2Cr_8O_{16}$ is unusual for two reasons: first the topological transition within an existing FM state and second that the topological nature depends on spin-lattice coupling and could therefore be manipulated by a field. The manuscript argues against a classic Peierls transition into the insulating phase. This is most certainly interesting from a basic science and future device point of view.

However, a few things badly need clarification. Cognizant of the fact that there is limited space and some things must necessarily be relegated to SM, it is still necessary to make clear in the main text what is being done. And, in the SM where there is plenty of room, it should be crystal clear.

First, it is not possible to tell which of the calculations are done spin-polarized ("magnetic"), if any. The supplemental material describes the paramagnetic tetragonal and paramagnetic monoclinic phases. As far as I can tell in the literature, there is no paramagnetic monoclinic phase since the monoclinic phase occurs fully within the FM state and remains after the MIT - a main interest point of this material. Were both of these phases calculated non-magnetically (un-spin-polarized) in DFT? If so, its very hard to draw any conclusions about the true band structure. The authors rigidly shift the bands to mimic the experimental gap, indicating that, again, this is within the FM state and should be calculated that way. Could exchange splitting itself contribute to the gap? Spin simply can't be ignored. And if it has not been ignored, it needs to be made far, far clearer what is happening in the calculations.

The SM mentions noncollinear calculations which indicate that maybe magnetism was, in fact, explored. This could, in principle, be used to resolve the in-plane magnetic orientation, but after the single instance in the methodology part, this concept is never mentioned again. Are there moments? Are they non-collinear? On a similar note, what *are* the calculated moments of the Cr atoms in the insulating phase? They should be integer (it is gapped), but this would require non-uniform spins. This would be a very interesting application of DFT but does not seem to have been done.

Again with respect to magnetism, the authors claim that the Peierl's mechanism is not operative because of the absence of a phonon soft mode at the CDW wavevector. But once again, the phonon calculations were done within the non-magnetic phase (by definition since methods to do it in the FM phase are currently lacking) and the transition of interest occurs in the FM phase. Given that there is certainly some and likely fairly strong spin-lattice coupling, calculations that lack spin-polarization do not reveal the true underlying physics.

Finally, the phase transition from tetragonal to monoclinic is initially described as "a structural distortion from a metallic tetragonal phase to an insulating monoclinic phase (Fig. 1) with $q_{CDW} = (1/2, 1/2, 0)$ ". Fig. 1 shows a larger monoclinic phase that does indeed look to be based on a $q=(1/2, 1/2, 0)$ supercell. But later, when calculating the Z_4 invariant, the authors state "we confirm that the monoclinic phase is indeed a trivial insulator (i.e., Z_4 index is 0 in the absence of CDW". I dug through the SM and thought I found the answer: the authors found a magnetic space group that is monoclinic but not the

same as the monoclinic cell resulting from the CDW, but then this sentence is included "Note that for simplicity and consistency with the literature, we will refer to both the paramagnetic and FM states by the lattice type of their structural space group, i.e., as the tetragonal phases of K₂Cr₈O₁₆, even though the magnetic lattice of latter is actually monoclinic". Are there two monoclinic cells? One ferromagnetic and metallic and the other ferromagnetic and insulating, the latter being the CDW cell? If it is not the CDW (post-MIT) cell that is calculated to be non-trivial, how can we understand that this is truly a topological transition?

I hope that these problems can be solved with better wording, but if all the calculations were done non-magnetically, I fear they are not useful for elucidating the interesting physics of this material.

In conclusion, much clarification is needed to justify the computational portion of this manuscript before publication is warranted.

Reviewer #2

(Remarks to the Author)

This manuscript reports the combined experimental and computational study of the hollandite compound K₂Cr₈O₁₆ that shows a metal-insulator transition within the ferromagnetically ordered phase. The authors revisit the mechanism of this transition and rule out the hitherto widely accepted Peierls scenario. They argue that the transition is topological in nature and driven by a nesting between the Weyl points.

This work makes at least two advances in understanding K₂Cr₈O₁₆. First, it uses inelastic neutron scattering to identify "chimneys" as the magnetic building blocks of the material. Second, it shows the absence of phonon softening near the transition, thus ruling out the conventional Peierls mechanism. K₂Cr₈O₁₆ is an important model system for a metal-insulator transition in the presence of magnetic order. Moreover, the idea of a metal-insulator transition driven by topological features of the band structure is certainly interesting, even if not thoroughly proved in the present version of the manuscript. Therefore, I am generally supportive of publication in Nature Comm. but only if stronger evidence for the occurrence of the Weyl points and their role in the transition could be demonstrated.

1. Leading magnetic interactions and the absence of phonon softening are convincingly proved experimentally. By contrast, the discussion of the Weyl points and their nesting fully relies on DFT and lacks any direct support from the experiment. K₂Cr₈O₁₆ is probably a correlated material that may not be fully described by DFT or even by DFT+U. In this context, ARPES measurements would be ideal in order to pinpoint the Weyl points. Alternatively, the dependence of the proposed Weyl points on the 'U' parameter should be discussed in detail. Are the appearance of these Weyl points near the Fermi level and their nesting robust features that do not depend on the computational parameters? Fig. 3c,d shows band structures with the Weyl points, but it is not quite clear whether those are calculated with or without 'U'. This information should be available directly on the graphs or in the figure caption, and the dependence on 'U' should be analyzed.

2. While the absence of phonon softening indicates an electronic origin of the transition, it is not quite clear why this transition is attributed to the nesting of the Weyl points and not to some other features of the electronic structure. A Fermi surface nesting would be a more natural scenario in this case. This possibility should be explored in detail, e.g., by calculating Lindhard susceptibility. On general grounds, it is not quite plausible that some features lying 0.1 eV away from the Fermi level trigger such a major transformation in the material.

3. It is highly desirable to pin down the magnetic moment direction, because it determines the exact positions of the Weyl points and the separation between them. This can be done by angle-dependent magnetization measurements on a single crystal.

4. While the metal-insulator transition in K₂Cr₈O₁₆ is dubbed topological, there seems to be no direct evidence for its topological nature. The formation of an axion insulator in the low-temperature monoclinic phase is speculated but never proved. In my opinion, the claims of the topological transition and axion-insulator phase are exaggerated and should be removed unless they can be directly supported by DFT calculations or experiments.

5. Previous studies have shown that the tetragonal structure of K₂Cr₈O₁₆ is stable in PBE, whereas imaginary phonons appear in PBE+U [Phys. Rev. B 90, 045124 (2014)]. Therefore, a phonon-driven transition becomes possible when correlations are sufficiently strong. While it does not seem to be the case experimentally, the agreement between the experimental and calculated phonon frequencies (Fig. 4e) is mediocre. It is not quite clear whether DFT or DFT+U was used in the phonon calculations, and it will be desirable to compare the experimental phonon spectrum with both calculations. Which calculation allows a better agreement with the experiment?

6. The INS fits are not very convincing because there is very little contrast in the experimental data. Please, show the constant-energy cuts along with the spin-wave fits in order to confirm the accuracy of the proposed model.

7. Calculated magnetic interactions are proportional to t^2/U , but the reasons for it are not explained in detail. The t^2/U dependence is typical for antiferromagnetic superexchange, but J₁-J₃ are all ferromagnetic. Which hopping parameters are considered in this case? I could imagine that FM superexchange follows the t^2/U dependence if 't' is the t_{2g}-e_g hopping, but is it really the case? What about the t_{2g}-t_{2g} hoppings? They must lead to antiferromagnetic superexchange interactions.

8. Ref. [19] is not displayed in the manuscript. I presume it is New J. Phys. (2020)

Reviewer #3

(Remarks to the Author)

$K_2Cr_3O_{16}$ ferromagnetic hollandite has been studied for almost two decades for presenting the prototype ferromagnetic metal-insulator transition, a rare phenomenon. In two seminal PRLs in 2009 and 2011 (cited as refs. 16, 17 in the manuscript), and also in another paper the year after (ref. 18), the following description was given:

$K_2Cr_3O_{16}$ was determined to be a quasi-1D system with Peierls distortion in the ferromagnetic insulating state. Peierls here is due to the dimerization of tetramers within the unit cell with no doubling of the lattice, but it does not bring along charge ordering on the Cr sites.

Ab initio calculations describe a double exchange mechanism between double chains via corner-shared oxygens, with exchange constants of 5.3 meV within the double chains but 10.6 meV between the double chains.

This results in a Fermi surface of strongly quasi-1D character with a nesting vector of (0,0,1) along the c direction.

The low-temperature structure was described in the $P112_1/a$ space group. In general, there is a mixed Cr^{3+}/Cr^{4+} valence. This picture has become well accepted.

A decade later, two ab initio studies—one cited as ref. 22 from NJP in 2020, the other from the same group in PRL the year before, but for some reason not cited (the manuscript refers to ref. 19 for a similar study, but it is actually missing from the reference list, probably a typo) — already described Weyl fermions in the system.

The present manuscript describes a different picture for the MIT. It presents experimental arguments against a Peierls instability and argues that the Weyl fermions bring along a correlation-induced topological phase transition that results in the MIT.

The low-temperature magnetic structure is described in the $C2'/m'$ space group, surprisingly, since it doesn't seem to be the best choice for ferromagnetic ordering.

The manuscript also argues against the interaction being double exchange, based on calculations of the dependence of the exchange on the hopping integral, and instead argues for a uniform $Cr^{3.75+}$ valence state with superexchange-like interactions between them.

Yet, the manuscript proposes exchange values of 6 and 10 meV, quite similar to those from 2011.

Therefore, the current manuscript proposes a drastically different chemical situation from the original picture and a dramatically different origin for the MIT.

A particularly glaring omission in the manuscript is that it strongly argues for the much improved quality of the samples, based on a modified high-pressure synthesis, but does not mention a single word about the modification. The original PRL from 2009 describes the high-pressure synthesis conditions in detail.

The manuscript proposes to set up a strong and interesting polemic in the literature.

The pros and cons should be more clearly and antagonistically described; the arguments in the current text are quite scattered.

However, and more importantly, there are both experimental and ab initio arguments in the current study.

The ab initio results don't seem to be qualitatively different from the 2019–20 results.

The experimental results of the inelastic X-ray scattering show a lack of phonon condensation.

This is employed as an argument against the Peierls instability.

However, the Peierls distortion proposed in 2011 is quite unusual.

Would it necessarily be accompanied by drastic changes in the phonon spectrum?

The inelastic neutron scattering shows a lack of drastic change in the spin-wave excitations across the MIT.

It is unclear—although the text makes it look as such—how this relates to the lack of a Peierls instability.

Version 1:

Reviewer comments:

Reviewer #1

(Remarks to the Author)

I am very pleased to report that all of my concerns were indeed addressed by the authors of this manuscript. My primary concern was that wording including "paramagnetic" was included frequently and I had thought it was being used to describe calculations that were non-magnetic. As it turns out, there are various crystallographic and magnetic phases and two of them are monoclinic and experimentally there are paramagnetic phases, but all calculations were magnetic. Likely, having performed all of this work, the authors understood what they were referring to and where throughout the manuscript, but from an outsider's point of view, it was quite confusing. The authors have taken the time to extensively explain what has been done in their response to my report and then, crucially, have included relevant wording in the manuscript and SM.

I can now endorse publication of this work as it is indeed interesting and will surely have an impact on the community.

Reviewer #2

(Remarks to the Author)

Most of my previous concerns have been addressed in the revised version. However, I have some additional queries that are inspired by the clarified computational methodology:

1. Both abstract and introduction put a lot of weight onto the correlated electron physics, but I am tempted to ask what the manifestations of the correlated physics are. In fact, the authors argue that the metallic phase of $K_2Cr_8O_{16}$ is best described using $U=0$, so they essentially conclude on its uncorrelated nature. One especially odd statement is the claim of the "correlation-induced topological phase transition" in the abstract, while the rest of the manuscript never mentions correlations at all. In my opinion, the claim of an MIT driven by topology instead of correlations would be a much better framework for this study, at least in its current form (but see #2 below). I would also advise the authors to completely re-write the abstract and highlight the concrete experimental findings, such as the absence of phonon softening, instead of the vague and arbitrary concepts like "quantum devices", which do not occur in this study at all.
2. On a related note, the authors argue that the INS spectra do not change across the MIT, so the exchange couplings must be similar in the metallic and insulating phases. This is indeed the case in Fig. S9b,c if the $U=0$ results are compared, but it would then indicate that the insulating phase must be uncorrelated too. On a closer look, the spin-wave fits are done with $J_1=6$ meV and $J_3=10$ meV, the values that seemingly correspond to $U=2$ eV in Fig. S9b. In contrast, the exchange couplings of up to 40 meV, as shown in Fig. S9a, should be inconsistent with the experiment. All of this is quite confusing and in fact casts doubt on the idea that the metallic phase is uncorrelated. Would not it be better to apply the same U -value to the metallic and insulating phases, considering that their exchange couplings are similar?
3. A comparison of the band structures calculated with different ' U ' is shown in Fig. 3 of the response letter, but it does not appear anywhere in the manuscript. In my opinion, it should be included. I am also confused by the panel 'e' of that figure. What does it show exactly? What is the color scheme?
4. I would still recommend updating the captions of Figs. 3 and S7 and writing explicitly the U -values used in each calculation. This will facilitate the reading.
5. It feels a bit strange that the magnetic couplings are in the range of 10 meV, but the INS data are mostly shown up to 4 meV only. The data obtained with the incident energy of 97 meV are also presented in Fig. S4, but without any comments or analysis, although these data clearly show more features than the low-energy data (note, for example, the flat band around 40 meV). Are the high-energy INS data equally well described by the spin model?
6. Fig. S5 is hard to read because of the many black lines that look more like "spaghetti's". One could use different shades of gray for each line, and the values of J_2 should be given either in the graph or in the figure caption.

Version 2:

Reviewer comments:

Reviewer #2

(Remarks to the Author)

My previous concerns have been addressed, and I gratefully recommend publication

Report of the first Referee

This manuscript is a combined computational and experimental work outlining a topological phase transition that occurs beneath (in temperature) a magnetic phase transition to a magnetic state. The compound, $K_2Cr_8O_{16}$ is unusual for two reasons: first the topological transition within an existing FM state and second that the topological nature depends on spin-lattice coupling and could therefore be manipulated by a field. The manuscript argues against a classic Peierls transition into the insulating phase. This is most certainly interesting from a basic science and future device point of view.

However, a few things badly need clarification. Cognizant of the fact that there is limited space and some things must necessarily be relegated to SM, it is still necessary to make clear in the main text what is being done. And, in the SM where there is plenty of room, it should be crystal clear.

Author Reply:

We thank the referee for the careful reading, constructive suggestions, and interest in our work. In the revised manuscript and Supplementary Material (SM), we have streamlined the language to improve clarity while maintaining technical accuracy. We now explicitly state the calculation configurations: All calculations were performed using spin-polarized settings.

Regarding the crystal structures, while the parent crystal structures are tetragonal (metallic) and monoclinic (insulating), the inclusion of ferromagnetic (FM) polarization breaks the tetragonal symmetry and results in a monoclinic symmetry. As a result, the system exhibits two distinct monoclinic configurations: one corresponding to a magnetic metallic state, and the other to a magnetic insulating state. While both share monoclinic symmetry, they are structurally and electronically distinct. We understand that our original phrasing may have caused confusion by not clearly distinguishing between crystallographic symmetry and symmetry breaking due to magnetic ordering. We have clarified the language throughout the manuscript and now explicitly indicate which symmetry/phase is being referred to in each case.

1. First, it is not possible to tell which of the calculations are done spin-polarized ("magnetic"), if any. The supplemental material describes the paramagnetic tetragonal and paramagnetic monoclinic phases. As far as I can tell in the literature, there is no paramagnetic monoclinic phase since the monoclinic phase occurs fully within the FM state and remains after the MIT - a main interest point of this material. Were both of these phases calculated non-magnetically (un-spin-polarized) in DFT? If so, its very hard to draw any conclusions about the true band structure. The authors rigidly shift the bands to mimic the experimental gap, indicating that, again, this is within the FM state and should be calculated that way. Could exchange splitting itself contribute to the gap? Spin simply can't be ignored. And if it has not been ignored, it needs to be made far, far clearer what is happening in the calculations.

Author Reply 1:

We thank the referee for the opportunity for clarification, and for raising the very important point highlighting the difference between a spin-polarized calculation and a paramagnetic calculation.

In the SM, we used the term *paramagnetic* in the context of our group theory analysis. It denotes the structural parent cell prior to any symmetry reduction imposed by spin polarization. We used it only as a crystallographic parent to extract symmetry relations. The “paramagnetic monoclinic” cell belongs to space group #14, while the paramagnetic tetragonal cell belongs to #87. These are subsequently reduced to the magnetic subgroup of #14.79 and #12.62 once the magnetic moments are taken into account. All first-principles calculations (band structures, J_{ij} , phonon, Weyl points analysis) are fully spin-polarized, for both the insulating and metallic phases.

Revisions

We understand that our original phrasing may have caused confusion by not clearly distinguishing between crystallographic and symmetry lowering due to magnetic ordering. In the revised manuscript and Supplementary Material, we have clarified the language and now explicitly state which symmetry is being referred to in each case.

We have also added the following clarifications in the SM Sec. S3A:

“All first-principles calculations (bandstructures, J_{ij} , phonon and Weyl points analyses) were performed in the ferromagnetic spin-polarized state, for both the insulating and metallic phases.”

We also added elaborations in the “Methods” Section of the Main Text:

"All DFT(+U)+SOC calculations for band structures, J_{ij} , phonon and Weyl points analyses were performed in the ferromagnetic spin-polarized state. "

2. The SM mentions noncollinear calculations which indicate that maybe magnetism was, in fact, explored. This could, in principle, be used to resolve the in-plane magnetic orientation, but after the single instance in the methodology part, this concept is never mentioned again. Are there moments? Are they non-collinear?

Author Reply 2:

We thank the referee for the opportunity for clarification. For the Weyl points calculations of the metallic phase, we performed noncollinear DFT+SOC calculations. The spin directions are unconstrained, but after self-consistency was achieved in the DFT calculation, a collinear configuration was found to be energetically most favorable. This configuration is also what was found from experiment (neutron diffraction). The calculated ferromagnetic moment of Cr in the metallic phase is $1.9 \mu_B$. The ordered moment was experimentally measured to be $1.6 \mu_B$ at 110 K. By extrapolating using the mean-field-type expression $m(T) = m_0(1 - T/T_C)^\beta$ with $\beta = 0.22$ and $T_C = 170$ K, we estimate the zero-temperature moment to be $m_0 \approx 2.0 \mu_B$, consistent with the calculated value.

In response to the referee's question, we systematically studied how variations in spin direction affect the nodal positions. Even though the exact positions of the Weyl points depend on the directions of the magnetic moment (as also reported by New Journal of Physics 22, 073062 (2020)), we found that the nodal points always form a cross-like feature about the nodal plane, such that nested nodal pairs are always present in the (0.5,0.5,0.0) and (-0.5,-0.5,0.0) directions of the tetragonal structure Brillouin zone (SM Fig. S8).

On a similar note, what *are* the calculated moments of the Cr atoms in the insulating phase? They should be integer (it is gapped), but this would require non-uniform spins. This would be a very interesting application of DFT but does not seem to have been done.

We thank the referee for raising this insightful question.

For the insulating monoclinic phase, we carried out fully noncollinear DFT+U+SOC calculations with unconstrained spin directions. After reaching self-consistency, the system relaxes back to a collinear ferromagnetic alignment, confirming that the collinear configuration is energetically preferred, consistent with neutron diffraction results.

In the metallic FM phase, each Cr ion carries a nominal charge of +3.75, corresponding to 2.25 electrons in its t_{2g} -like orbitals (d_{xy} , d_{xz} , d_{yz}). This fractional occupation ensures that the bands cannot be fully filled, resulting in a metallic or semi-metallic state.

In the insulating FM phase, two key structural changes allow for gap formation. First, the monoclinic structure distortion lifts the degeneracy between the d_{xz} and d_{yz} orbitals. Second, the four Cr atoms in each Cr_4O_6 chimney become crystallographically inequivalent, further splitting the d-orbital levels into twelve non-degenerate states. As a result, the nine electrons from four Cr atoms (4×2.25) can redistribute unevenly among the twelve orbitals. This redistribution enables nominal integer orbital fillings, which allows a band gap to open.

The calculated magnetic moment per Cr atom in the insulating phase is $2.1 \mu_B$, in close agreement with the experimental value of $2.2 \mu_B$ measured at 10 K. This is consistent with the nominal mixed-valence expectation of $2.25 \mu_B/\text{Cr}$: $[2\text{Cr}^{3+} (d^3) + 6\text{Cr}^{4+} (d^2)]/8$. The small reduction in the calculated value, compared with the ionic value, reflects Cr–O covalency and inter-site hybridization, which transfer part of the spin density onto the oxygen ligands and slightly delocalize it across Cr sites. Importantly, the total electron count per primitive cell remains integer, as required for an insulating state, and the total magnetic spin moment per unit cell is also an integer.

Revision:

We have updated the manuscript to include the following sentence regarding the moment size: "This is consistent with the calculated ($2.1 \mu_B$) and experimentally measured ($2.2 \mu_B$) Cr magnetic

moments. Moreover, the total electron count and the total spin moment per primitive cell are both integers, a condition necessary for the opening of a band gap.”

We have also included the following to the main text:

“In addition, we find that the positions of the nodal points are robust. Regardless of the direction of the magnetic moment, we found that the nodal points always form a cross-like feature about the nodal plane, such that nested nodal pairs are always present in the (0.5,0.5,0.0) and (-0.5,-0.5,0.0) directions of the tetragonal Brillouin zone (Figs. 3(b) and SM Fig. S8).”

Figure 1. Cross-like nodal points and their dependence on the directions of the in-plane magnetic moment. (a-h) Evolution of the nodal points (gapless points around the $k_c = \pm\pi/c$ plane) as a function of the magnetization direction. Here, the angle defines the direction of the magnetic moment relative to the a_M -lattice vector of the monoclinic supercell, which increases counterclockwise within the $a_M b_M$ -plane (as defined in Fig. 1(a) of Main Text). Although varying the direction of the magnetic moment shifts the positions of the nodal lines and position of the nodes, the overall cross-like feature persists, ensuring that nested nodal pairs persist along the directions of [0.5, 0.0, 0.0] and [0.0, 0.5, 0.0] in the monoclinic BZ (corresponding to [0.5, -0.5, 0.0] and [0.5, 0.5, 0.0] in the tetragonal BZ). Circle radii at the nodal points are proportional to $\log(\text{gap})$, while color intensity scales as $[\log(\text{gap})]^2$ using the colormap from (a), highlighting regions with smaller band gaps.

3. Again with respect to magnetism, the authors claim that the Peierl's mechanism is not operative because of the absence of a phonon soft mode at the CDW wavevector. But once again, the phonon calculations were done within the non-magnetic phase (by definition since methods to do it in the FM phase are currently lacking) and the transition of interest occurs in the FM phase. Given that there is certainly some and likely fairly strong spin-lattice coupling, calculations that lack spin-polarization do not reveal the true underlying physics.

Author Reply 3:

We thank the referee for raising this important point and for the opportunity to clarify the role of spin polarization in our phonon calculations.

Contrary to what may have been implied in our original submission, the phonon calculations were carried out in a spin-polarized state. All phonon dispersions reported in this work were obtained using the finite-displacement supercell method with spin-polarized DFT, based on Quantum ESPRESSO force calculations in the experimentally observed ferromagnetic configuration. This approach captures the spin-lattice coupling within the framework of DFT.

Specifically, for the metallic tetragonal phase, phonons were computed with the following parameters (see Supplementary S3D and Table S4):

- **Supercell:** $1 \times 1 \times 4$
- **Displacement amplitude:** 0.01 Å
- **k-point grid for forces:** shifted $2 \times 2 \times 2$
- **Hubbard U:** 0 eV
- **Spin configuration:** collinear ferromagnetic
- **Plane-wave cutoff:** 40 Ry
- **Method:** finite-displacement phonons using *Phonopy + Quantum ESPRESSO*

This calculation reveals no imaginary phonon modes, including at the CDW wavevector $q = (0.5, 0.5, 0)$. The results are directly compared with inelastic x-ray scattering measurements in Fig. 4e of the main text, showing good agreement without any anomalous softening near the transition.

In light of this, we respectfully clarify that the statement "by definition, methods to do [phonon calculations] in the FM phase are currently lacking" is not correct. In fact, two established first-principles frameworks allow phonon calculations in magnetically ordered phases:

1. The **finite-displacement method** (used here) is always compatible with spin-polarized states, since its forces are evaluated from fully self-consistent spin-polarized calculations.
2. **Density-functional perturbation theory (DFPT)** with spin polarization has been available since the early 2000s (e.g., Baroni *et al.*, Review of Modern Physics **73**, 515 (2001)), and is implemented in several DFT packages including Quantum ESPRESSO. Spin-polarized DFPT is routinely used in modern studies of magnetic materials, including phonon dispersions in bcc Fe and Fe-Co alloys (e.g., Nature Communications, **6**, 9961 (2015); Physical Review **B** 109, 184306 (2024)).

In summary, the spin-lattice coupling relevant to the FM state of $K_2Cr_8O_{16}$ is already captured by our finite-displacement calculations. The absence of a soft mode at the CDW wavevector in this fully spin-polarized framework rules out a conventional Peierls mechanism as the driver of the metal-insulator transition. This strengthens our conclusion that the transition arises from an electronic (topological) instability rather than a structural one.

Revision:

We have revised the Methods section of the main text to explicitly address this point. In addition, as part of our response to Question 1, we have added further elaboration in the "Methods" section of the main text:

"All DFT calculations for band structures, J_{ij} , phonon, Berry curvature and Weyl points analyses were performed in the ferromagnetic spin-polarized state."

4. Finally, the phase transition from tetragonal to monoclinic is initially described as "a structural distortion from a metallic tetragonal phase to an insulating monoclinic phase (Fig. 1) with $q_{CDW} = (1/2, 1/2, 0)$ ". Fig. 1 shows a larger monoclinic phase that does indeed look to be based on a $q=(1/2, 1/2, 0)$ supercell. But later, when calculating the Z_4 invariant, the authors state "we confirm that the monoclinic phase is indeed a trivial insulator (i.e., Z_4 index is 0 in the absence of CDW)".

I dug through the SM and thought I found the answer: the authors found a magnetic space group that is monoclinic but not the same as the monoclinic cell resulting from the CDW, but then this sentence is included "Note that for simplicity and consistency with the literature, we will refer to both the paramagnetic and FM states by the lattice type of their structural space group, i.e., as the tetragonal phases of $K_2Cr_8O_{16}$, even though the magnetic lattice of latter is actually monoclinic". Are there two monoclinic cells? One ferromagnetic and metallic and the other ferromagnetic and insulating, the latter being the CDW cell? If it is not the CDW (post-MIT) cell that is calculated to be non-trivial, how can we understand that this is truly a topological transition?

Author Reply 4:

We thank the referee for the careful reading of our manuscript. The referee is correct that there are two monoclinic magnetic cells: one corresponding to the ferromagnetic–metallic state (space group #12.62) and the other to the ferromagnetic–insulating state (space group #14). The underlying crystallographic parent structures, before including magnetic ordering, are metallic tetragonal (space group #87) and insulating monoclinic (space group #14). Once ferromagnetic order is imposed, the tetragonal parent cell is reduced to monoclinic symmetry. For our calculations, we employed the same supercell lattice—specifically, the unit cell of the insulating phase, which is a $(2 \times 2 \times 1)$ supercell for the metallic crystal structure. While both phases adopt a monoclinic lattice, they are not equivalent: the monoclinic lattices belong to different space groups, since atomic displacements in the insulating phase break additional symmetries and alter the atomic positions relative to the metallic phase.

In the statement, "we confirm that the monoclinic phase is indeed a trivial insulator (i.e., Z_4 index is 0 in the absence of CDW)," we are referring to the insulating monoclinic cell (post-MIT). Our calculations show that its band topology is trivial. Meanwhile, our Berry curvature and Weyl point analyses of the metallic phase (see Fig. 3b) confirm that its band topology in the magnetic monoclinic phase is non-trivial. The transition between these two phases coincides with the MIT, confirming that it is indeed a topological transition.

Revision:

We understand that this was not clear and we have revised the manuscript to address this point. In the revised manuscript and Supplementary Material, we have clarified the language and now explicitly state which symmetry is being referred to in each case. Specifically, we now reference

these phases as “metallic phase” or “insulating monoclinic phase” etc. to be more explicit. We have also added the following sentence to the main text. “The FM spin polarization reduces the symmetry from the paramagnetic space group $I4/m$ (#87) to the magnetic subgroup $C2'/m'$ (#12.62), which is monoclinic. “

5. I hope that these problems can be solved with better wording, but if all the calculations were done non-magnetically, I fear they are not useful for elucidating the interesting physics of this material.

In conclusion, much clarification is needed to justify the computational portion of this manuscript before publication is warranted.

Author Reply 5:

We are grateful to the referee for giving us this opportunity to clarify the details of the calculations. The comments have greatly helped improve the quality and clarity of our manuscript. We explicitly mention the correct symmetry/phase for clarity, and all calculations in this work were done spin polarised, which is now clearly stated in both the Main text and the SM.

Report of the second Referee

This manuscript reports the combined experimental and computational study of the hollandite compound $\text{K}_2\text{Cr}_8\text{O}_{16}$ that shows a metal-insulator transition within the ferromagnetically ordered phase. The authors revisit the mechanism of this transition and rule out the hitherto widely accepted Peierls scenario. They argue that the transition is topological in nature and driven by a nesting between the Weyl points.

This work makes at least two advances in understanding $\text{K}_2\text{Cr}_8\text{O}_{16}$. First, it uses inelastic neutron scattering to identify "chimneys" as the magnetic building blocks of the material. Second, it shows the absence of phonon softening near the transition, thus ruling out the conventional Peierls mechanism. $\text{K}_2\text{Cr}_8\text{O}_{16}$ is an important model system for a metal-insulator transition in the presence of magnetic order. Moreover, the idea of a metal-insulator transition driven by topological features of the band structure is certainly interesting, even if not thoroughly proved in the present version of the manuscript. Therefore, I am generally supportive of publication in Nature Comm. but only if stronger evidence for the occurrence of the Weyl points and their role in the transition could be demonstrated.

Author Reply:

We appreciate the referee's thoughtful assessment and their general support for the publication of this work in Nature Communications. To address the referee's primary concern — the presence, location, and role of the Weyl points — we have performed angle-dependent magnetization measurements to precisely determine the direction of the magnetic order. Based on these results, we recalculated the band structure, incorporating the refined spin orientation. The updated band structure reveals Weyl points connected by a nesting vector of approximately $(0.75, 0.75, 0)$, which, while not exactly commensurate, closely aligns with the experimentally observed lattice distortion vector $(0.5, 0.5, 0)$. In addition, we find that the positions of the nodal points are robust. Regardless of the direction of the magnetic moment, we found that the nodal points always form a cross-like feature about the nodal plane, such that nested nodal pairs are always present in the $(0.5, 0.5, 0.0)$ and $(-0.5, -0.5, 0.0)$ directions of the tetragonal Brillouin zone. Importantly, aside from these nodal points, there are no indications of conventional Fermi surface nesting or other electronic instabilities along this direction. This makes the correlation between the Weyl point nesting and the structural distortion particularly compelling.

Additionally, we now include preliminary ARPES data in this response letter to demonstrate consistency between the calculated band structure and experimental observations. These data suggest a Weyl point nesting vector of approximately $(0.48, y, 0.02)$, which is consistent with the lattice distortion vector, although the y -component cannot be precisely determined at this stage. We hope this can be further clarified in a follow-up study, which would require a careful photon energy scan.

We wish to however emphasize that while our results show that the metal-insulator transition is a topological phase transition within a magnetically ordered state — connecting a Weyl semimetal and a topologically trivial insulating state — this study does not directly demonstrate that the Weyl points drive the system into an axion-insulating ground state. While the compound

is certainly a good candidate, we have intentionally left this as an open question for future research. In the revised manuscript, we have clarified this point further. For example, it is mentioned that $K_2Cr_8O_{16}$ is exhibiting “potential axionic properties”.

We thank the referee for the opportunity to clarify these issues and for their valuable input, which has helped us strengthen the manuscript.

1. Leading magnetic interactions and the absence of phonon softening are convincingly proved experimentally. By contrast, the discussion of the Weyl points and their nesting fully relies on DFT and lacks any direct support from the experiment. $K_2Cr_8O_{16}$ is probably a correlated material that may not be fully described by DFT or even by DFT+U. In this context, ARPES measurements would be ideal in order to pinpoint the Weyl points. Alternatively, the dependence of the proposed Weyl points on the 'U' parameter should be discussed in detail. Are the appearance of these Weyl points near the Fermi level and their nesting robust features that do not depend on the computational parameters? Fig. 3c,d shows band structures with the Weyl points, but it is not quite clear whether those are calculated with or without 'U'. This information should be available directly on the graphs or in the figure caption, and the dependence on 'U' should be analyzed.

Author Reply 1:

We thank the referee for this important and insightful point. As the referee correctly notes, the degree of electronic correlation in $K_2Cr_8O_{16}$ remains an open question. While there is experimental evidence suggesting moderate correlation effects—such as those indicated by HAXPES measurements [Physical Review X 5, 041004 (2015)]—the extent to which standard DFT or DFT+U captures these correlations remains an area of active discussion.

Nonetheless, we find that standard DFT (GGA) performs remarkably well in describing several key properties of $K_2Cr_8O_{16}$, supporting its use in this context:

- **Magnetic interactions:** The exchange interactions calculated via DFT are consistent with our neutron scattering results, showing that J_1 and J_2 (nearest-neighbor couplings) are smaller than J_3 (the third-nearest-neighbor coupling). This agreement indicates that DFT captures the essential magnetic physics observed experimentally.
- **Curie temperature:** While DFT predicts a Curie temperature of ~ 300 K (Fig. 2 below)—higher than the experimental value of ~ 170 K—this level of overestimation is not unexpected. It is well known that DFT, particularly in the mean-field treatment of magnetism, often yields elevated T_C values. Importantly, the qualitative agreement (ferromagnetic ordering and scale of magnetic interactions) supports the reliability of the magnetic picture obtained.
- **Moment size:** The calculated ferromagnetically ordered magnetic moment of the Cr of insulating monoclinic phase is $2.1 \mu_B$ (Fig. 2 below) which is close to the experimental value of $2.2 \mu_B$.

- **Band structure:** The DFT band structure agrees well with our preliminary ARPES data presented below (Fig. 4), further validating the reliability of DFT in capturing the low-energy electronic states relevant to the Weyl physics.

While we agree that a full treatment of dynamic correlations (e.g., via DMFT or GW) would be necessary to describe certain high-energy features or the Mott physics, the physical phenomena discussed in this work—namely, the topology of the metallic state and its relation to the observed structural distortion—are predominantly governed by the symmetry and geometry of the band structure, which are already well captured by DFT. We believe, therefore, that DFT provides a reliable framework for identifying and analyzing the Weyl points and their nesting in $K_2Cr_8O_{16}$.

Figure 2. Calculated magnetization as a function of temperature. See Fig. 1 in Ref. Nakao, A. et al. "Observation of Structural Change in the Novel Ferromagnetic Metal–Insulator Transition of $K_2Cr_8O_{16}$." *J. Phys. Soc. Jpn.* 81, 054710 (2012) for measured magnetization as a function of temperature.

Robustness of Weyl points and role of U:

The Weyl points we report are consistent with prior DFT studies [e.g., *New Journal of Physics* **22**, 073062 (2020)] and remain robust under small variations of magnetic configuration and exchange-correlation functionals. To directly address the referee’s concern, we have performed additional PBE+U calculations with U ranging from 1 eV to 4 eV (Fig. 3). We found that other than the slight increase in the bandwidths, the qualitative features of electronic band structure remains largely the same:

- The overall electronic structure remains qualitatively similar across this range.
- Certain segments of the nodal plane begin to gap, while the cross-shaped nodal features and their corresponding nesting vectors near $(\pm 0.5, \pm 0.5, 0)$ are preserved for $U \leq 3$ eV.
- Some partial gapping appears for $U = 2 - 3$ eV, but the essential topology and nesting behavior remain intact.
- Only for $U > 4$ eV does strong localization lead to a complete gapping of the nodal features.

These results confirm that the Weyl point features and their nesting are robust over a broad and physically reasonable range of U .

Figure 3. Wannier bandstructure of $K_2Cr_8O_{16}$ for (a) $U = 0\text{eV}$, (b) $U = 1\text{eV}$, (c) $U = 2\text{eV}$ and (d) $U = 3\text{eV}$. (e) The positions of the nodal points for Hubbard U for $0 < U < 3\text{eV}$. The cross-like feature remains robust even under the influence of Hubbard U for $0 < U < 3\text{eV}$.

Justification of $U = 0$:

With this in mind, we employed DFT-PBE ($U = 0$) for the metallic phase in this work. This choice is both standard practice and physically well justified in the context of strongly correlated materials exhibiting a metal-insulator transition, such as VO_2 and V_2O_3 [Physics Review Research **3**, 033286 (2021); Science **362**, Issue 6418 pp. 1037-1040 (2021); The Journal of Chemical Physics **160**, 134101 (2024)]. In metallic systems, the d-electrons are itinerant and the Coulomb interaction is strongly screened by conduction electrons. Consequently, the effective on-site interaction is significantly reduced, and the electronic properties of the metallic phase are already well captured by conventional DFT. As noted in a recent article [Science **362**, Issue 6418 pp. 1037-1040 (2021)], the metallic rutile phase of VO_2 is described as having “little electronic correlation,” in contrast to the correlated insulating monoclinic phase.

Applying a static Hubbard U in a metallic phase can introduce unphysical artifacts. Several studies have shown that it can over-localize the electrons, artificially open a gap, or induce incorrect magnetic ordering [The Journal of Chemical Physics **160**, 134101 (2024)]. Similarly, hybrid functionals, which introduce a screened Fock exchange equivalent to a static U , have been shown to spuriously drive the metallic phase into an insulating state or cause it to relax to the low-temperature phase [Physical Review B **86**, 081101(R) (2012); Journal of Computational Chemistry **41**, 1 (2019); Physical Review B **103**, 075134 (2021)]. These results caution against using DFT+ U indiscriminately in metallic regimes. Theoretically, the Hubbard U in DFT+ U is a static approximation to the frequency-dependent self-energy. A more accurate description of the

dynamically screened Coulomb interaction in the metallic state requires a many-body approach such as GW or DMFT [Physical Review Research **3**, 033286 (2021)]. However, these methods are beyond the scope of the present study, which focuses on the low-energy band topology accessible within static mean-field theory.

Given that the focus of this work is on the topology of the metallic state, which is largely governed by symmetry and geometry of the band structure, DFT without U offers a physically justified and computationally stable framework.

ARPES measurements:

Following the referee's suggestion to provide experimental support for the Weyl points, we have conducted initial ARPES measurements on $K_2Cr_8O_{16}$. Identifying the exact location of the Weyl points would have been most straightforward through band structure measurements on the (0,0,1) surface as a function of k_z . However, due to the quasi-one-dimensional crystal structure of $K_2Cr_8O_{16}$, the material cleaves preferentially along the {0,1,0} planes, which limits direct access to the ideal geometry. Moreover, the quasi-1D nature of the compound leads to an exponential suppression of photoemission intensity near the Fermi level, further constraining the measurements.

Despite these limitations, we are able to partially reconstruct the electronic structure by probing the (k_x, k_z) plane (see Fig. 5). In this geometry, we observe Fermi surface features that is consistent with DFT predictions. A clear nesting feature is manifested along (0,0,1), which is seen in our and earlier DFT work [PRL 107, 266402 (2011)] (although, as known, the structure distortion is along (0.5, 0.5, 0)). Using circularly polarized light (left and right), we selectively probe band crossings with different chiralities. These measurements reveal intensity modulations, which are switched on and off depending on the polarization, at the expected Weyl point locations, in agreement with the predicted chiral band structure.

Taken together with symmetry considerations, the observed patterns are consistent with the presence of nested Weyl points with approximate nesting vectors of $\approx (0.48, y, 0.02)$ and $(0.44, y, 0)$, in agreement with the DFT results. While the limited data prevent precise determination of the y -component of the nesting vector, the overall experimental observations support the theoretical picture of Weyl points and their nesting along (0.5, 0.5, 0). Further ARPES measurements, in particular a more detailed photon energy scan, will be necessary to confirm the y component of the nesting vector more definitively.

Figure 5. (a, b) Measured Fermi surface of $K_2Cr_8O_{16}$ in the k_x - k_z plane using circularly right (CR) and circularly left (CL) polarised light, respectively. The dashed box indicates the Brillouin zone. The dark spot on the right side is an artifact caused by the analyzer deflector.

Revision:

We have updated the figures to directly include the value of U used. Moreover, we added in the Methods section of the Main Text:

“DFT calculations were carried out with $U = 4.0$ eV for the insulating monoclinic phase to capture electron-correlation effects, whereas $U = 0$ eV was used for the metallic phase, where metallic screening is expected to strongly reduce the effective on-site interactions.”

2. While the absence of phonon softening indicates an electronic origin of the transition, it is not quite clear why this transition is attributed to the nesting of the Weyl points and not to some other features of the electronic structure. A Fermi surface nesting would be a more natural scenario in this case. This possibility should be explored in detail, e.g., by calculating Lindhard susceptibility. On general grounds, it is not quite plausible that some features lying 0.1 eV away from the Fermi level trigger such a major transformation in the material.

Author Reply 2:

We thank the referee for this insightful comment. Indeed, a conventional Fermi surface nesting at the vector $(0.5, 0.5, 0)$ is absent in our and previously calculated Fermi surfaces (e.g. Fig. 6 in Physical Review B **80**, 024416 (2009)). While a 3D-like pocket near the center of the Brillouin zone appears in the calculations (but only very weakly in the experimental data; Figure 5, reply 1), such a 3D structure is generally difficult to nest effectively.

Early works [Physical Review Letters **107**, 266402 (2011)] presents calculated Fermi surface (comparable to our calculations) with a nesting feature along $(0, 0, 1)$, which was the foundation behind the suggestion of a Peierls transition. However, crucially, the observed crystal distortion is along $(0.5, 0.5, 0)$ [Physical Review Letters **107**, 266402 (2011)], which is perpendicular to $(0, 0, 1)$. Thus, the nesting along $(0, 0, 1)$ does not drive the lattice modulation. Instead, our key finding is that Weyl points are located near $(0.5, 0.5, 0)$, which coincides with the experimentally observed lattice distortion vector. While this suggests a possible connection between Weyl point

nesting and the transition, we acknowledge that direct evidence for the Weyl points driving the distortion is currently lacking. We have softened the language accordingly.

Lindhard susceptibility:

We agree that an explicit calculation of the Lindhard susceptibility $\chi_0(\mathbf{q}, \omega)$ could be informative. As the referee may know, we respectfully note that the phonon spectrum already incorporates this quantity (see Review of Modern Physics **89**, 015003 (2017), Giustino). Below we explain how the phonon spectrum already embeds the relevant electronic response, and why we do not expect a qualitative change in the Lindhard susceptibility when U is varied moderately.

In short, the phonon spectrum incorporates the full dielectric screening, extending beyond the Lindhard function and even beyond the random phase approximation (RPA). Thus, the phonon dispersion indirectly mirrors the behavior of the noninteracting Lindhard susceptibility, χ_0 : an enhancement of $\chi_0(\mathbf{q}, 0)$ (and its dressed χ) reduces $\epsilon(\mathbf{q}, 0)$, increases the magnitude of $C^{\text{el}}(\mathbf{q})$, which is the electronic screening contribution to the dynamical matrix, and results in a phonon softening at the corresponding wavevector.

To begin, we introduce the (noninteracting) Lindhard susceptibility that controls the basic electron-hole response.

$$\chi_0(\mathbf{q}, \omega) = \frac{2}{N_k} \sum_{nm\mathbf{k}} \frac{f_{n\mathbf{k}} - f_{m,\mathbf{k}+\mathbf{q}}}{\epsilon_{n\mathbf{k}} - \epsilon_{m,\mathbf{k}+\mathbf{q}} + \omega + i\eta} |\langle n\mathbf{k} | e^{i\mathbf{q}\cdot\mathbf{r}} | m, \mathbf{k} + \mathbf{q} \rangle|^2 \quad (1)$$

Here N_k is the number of \mathbf{k} points, $f_{n\mathbf{k}}$ are Fermi occupations, $\epsilon_{n\mathbf{k}}$ are band energies, η is a positive infinitesimal, and the matrix element couples Bloch states at \mathbf{k} and $\mathbf{k}+\mathbf{q}$.

Including Hartree screening yields the RPA response:

$$\chi_{\text{RPA}}(\mathbf{q}, \omega) = \frac{\chi_0(\mathbf{q}, \omega)}{\mathbf{1} - v(\mathbf{q})\chi_0(\mathbf{q}, \omega)}.$$

Here $v(\mathbf{q})$ is the bare Coulomb kernel. DFT further improves this by adding the exchange-correlation kernel f_{xc} , giving:

$$\chi(\mathbf{q}, \omega) = \frac{\chi_0(\mathbf{q}, \omega)}{\mathbf{1} - [v(\mathbf{q}) + f_{\text{xc}}(\mathbf{q}, \omega)]\chi_0(\mathbf{q}, \omega)}.$$

which is precisely the level that DFPT phonons probe in the static limit.

In DFPT, phonon frequencies are the eigenvalues of the dynamical matrix, which contains an electronic term built from the screened density response χ . This term makes the connection between dielectric screening and lattice dynamics explicit.

$$D_{\kappa\alpha,\kappa'\beta}(\mathbf{q}) = \frac{1}{\sqrt{M_\kappa M_{\kappa'}}} C_{\kappa\alpha,\kappa'\beta}(\mathbf{q}),$$

$$C(\mathbf{q}) = C^{\text{ion}}(\mathbf{q}) + C^{\text{el}}(\mathbf{q}).$$

Here D is the dynamical matrix, C are the Fourier-transformed interatomic force constants, M_κ is ionic mass, C^{ion} is the bare ionic contribution, and C^{el} is the electronic screening contribution.

The electronic part can be written as a bilinear form of the screened response χ with respect to the linear variation of the self-consistent potential induced by ionic displacements:

$$C_{\kappa\alpha,\kappa'\beta}^{\text{el}}(\mathbf{q}) = \int d\mathbf{r} \int d\mathbf{r}' g_{\kappa\alpha}(\mathbf{r}; \mathbf{q}) \chi(\mathbf{r}, \mathbf{r}'; \mathbf{q}, \omega = 0) g_{\kappa'\beta}(\mathbf{r}'; -\mathbf{q}),$$

where $\chi(\omega=0)$ is the static screened response that includes v and f_{xc} , and the integrals run over real space. Here, $g_{\kappa\alpha}$ is the electron-phonon perturbation that follows from the linear variation of the self-consistent Kohn-Sham potential, which itself depends on the variation of the self-consistent potential via,

$$g_{\kappa\alpha}(\mathbf{r}; \mathbf{q}) = \frac{\delta V_{\text{SCF}}(\mathbf{r})}{\delta u_{\kappa\alpha}(\mathbf{q})},$$

$$\delta V_{\text{SCF}}(\mathbf{r}) = \int d\mathbf{r}' [v_c(\mathbf{r}, \mathbf{r}') + f_{xc}(\mathbf{r}, \mathbf{r}')] \delta n(\mathbf{r}'),$$

where V_{SCF} is the self-consistent potential and δn is the induced charge density. We see that the same Hartree term v_c and the exchange-correlation kernel f_{xc} that dress χ_0 and define χ also appear in the electron-phonon vertex.

Thus, the same Hartree and exchange-correlation kernels that dress χ_0 appear directly in both χ and $g_{\kappa\alpha}$, establishing that C^{el} samples the fully screened χ rather than the bare χ_0 . In the finite-displacement supercell method, the same screening physics is captured implicitly, since forces are computed self-consistently with spin-polarized DFT. Forces are taken from self-consistent DFT calculations for small displacements, so δn and δV_{SCF} are obtained by solving the Kohn-Sham response with the same Hartree and exchange-correlation kernels. The resulting force-constant matrix C recovers the same C^{el} as DFPT in the static limit.

These relations show that if a strong nesting or divergent electronic response existed at a given q , $\epsilon(q,0)$ would be reduced and the electronic contribution $C^{\text{el}}(q)$ would become strongly negative, which would appear as a softening or an imaginary phonon at that q . Our spin-polarized phonon dispersions show no softening at $q = (0.5, 0.5, 0)$. This disfavors a conventional Fermi-surface-driven Peierls mechanism.

Finally, Eq. (1) highlights that χ_0 depends on band energies and matrix elements. The band structures in Fig. S3a-d show that increasing U mainly narrows the bandwidth while keeping the near-Fermi topology similar. The phase space for electron-hole excitations and the matrix-

element structure therefore remain comparable, so $\chi_0(q,0)$ should not change qualitatively for moderate U. No new sharp nesting instabilities are expected to arise from bandwidth renormalization alone.

3. It is highly desirable to pin down the magnetic moment direction, because it determines the exact positions of the Weyl points and the separation between them. This can be done by angle-dependent magnetization measurements on a single crystal.

Author Reply 3:

We thank the referee for the valuable suggestions. Following this, we have performed angle-dependent magnetization measurements on single crystal $K_2Cr_8O_{16}$. Specifically, we measured the magnetization as a function of applied magnetic field at various angles within the ab plane. These measurements indicate that the magnetic moments are oriented approximately 22.5° away from the a/b crystallographic axis. We have included these new results in the main text (Fig. 1(c) in main text) and of the revised manuscript.

In response to the referee's inquiry about the impact of moment direction on Weyl points, we conducted a systematic theoretical study of how varying the spin orientation affects the positions of the Weyl nodes. Consistent with prior reports (New Journal of Physics **22**, 073062 (2020)), the exact Weyl point locations shift with the magnetic moment direction. However, importantly, the nodal points consistently form a characteristic cross-like structure on the nodal plane, with nested pairs robustly present along the $\{\pm 0.5, \pm 0.5, 0.0\}$ directions in the tetragonal Brillouin zone (see Figs. 3b and SM Fig. S8).

Revision:

In the Main Text, we added the following line:

“In addition, we find that the positions of the nodal points are robust. Regardless of the direction of the magnetic moment, we found that the nodal points always form a cross-like feature about the nodal plane, such that nested nodal pairs are always present in the (0.5,0.5,0.0) and (-0.5,-0.5,0.0) directions of the tetragonal Brillouin zone (Fig. 3(b) and SM Fig. S8)”

4. While the metal-insulator transition in $K_2Cr_8O_{16}$ is dubbed topological, there seems to be no direct evidence for its topological nature. The formation of an axion insulator in the low-temperature monoclinic phase is speculated but never proved. In my opinion, the claims of the topological transition and axion-insulator phase are exaggerated and should be removed unless they can be directly supported by DFT calculations or experiments.

Author Reply 4:

We agree with the referee that firm claims of a topological transition to an axion-insulator phase would be exaggerated. Our main claim, however, is that the MIT is a topological phase transition as we have shown that the metallic phase is a Weyl semi metal (Fig. 3(b)), as is also known in and highly reproducible from literature [New Journal of Physics. **22**, 073062 (2020)] and that the

insulating phase is *at least* topologically trivial. A transition between these two phases with distinct topological character naturally qualifies as a topological phase transition.

In addition, our work provides novel evidence that the Weyl points of opposite chirality are nested by wavevectors $q = [0.75, 0.75, 0.0]$, which correspond closely to distortion lattice wavevectors observed experimentally via single-crystal X-ray diffraction [Physical Review Letters **107**, 266402 (2011); Journal of the Physical Society of Japan **81**, 054710 (2012)]. This nesting offers a plausible microscopic mechanism for the MIT linked to electronic topology. However, a nesting of Weyl points do not guarantee an axion insulating ground state. Importantly, throughout the manuscript and abstract, we have carefully phrased the possibility of an axion insulating state only as a potential scenario or “possibly axionic in nature,” without making conclusive claims. We have now further clarified in the revised manuscript that the axion insulator is one of several candidate ground states, and that future theoretical and experimental studies are necessary to confirm or refute this.

We hope this clarifies that our primary focus remains on demonstrating the topological character of the MIT, without overstating the evidence for an axion insulating phase.

5. Previous studies have shown that the tetragonal structure of $K_2Cr_8O_{16}$ is stable in PBE, whereas imaginary phonons appear in PBE+U [Phys. Rev. B **90**, 045124 (2014)]. Therefore, a phonon-driven transition becomes possible when correlations are sufficiently strong. While it does not seem to be the case experimentally, the agreement between the experimental and calculated phonon frequencies (Fig. 4e) is mediocre. It is not quite clear whether DFT or DFT+U was used in the phonon calculations, and it will be desirable to compare the experimental phonon spectrum with both calculations. Which calculation allows a better agreement with the experiment?

Author Reply 5:

We thank the referee for highlighting the excellent phonon study carried out using DFT+U, as reported in Ref. [Physical Review B **90**, 045124 (2014)]. We clarify that the phonon calculations shown in Fig. 4e were performed using DFT-PBE with $U = 0$ eV. As also outlined in reply 1, this choice was made for both physical and practical reasons:

1. **Physical justification:** In the high-temperature metallic phase of $K_2Cr_8O_{16}$, the electrons are itinerant and screening is strong, effectively reducing the impact of local Coulomb interactions. Applying a static U in this regime without proper justification may often introduces unphysical artifacts, such as spurious gap openings or magnetic order, as discussed in the context of similar correlated metals [Physical Review Research **3**, 033286 (2021); Science **362**, Issue 6418 pp. 1037-1040 (2021); The Journal of Chemical Physics **160**, 134101 (2024)].
2. **Consistency with experiment:** The experimental phonon spectrum does not show any soft modes or imaginary frequencies that would indicate a phonon-driven instability. In contrast, Ref. [Physical Review B **90**, 045124 (2014)] shows that including a Hubbard U

can introduce such imaginary modes in the tetragonal phase — a result that appears to conflict with our experimental findings.

Based on these considerations, we chose to rely on DFT ($U = 0$), which gives a phonon spectrum that, while not in perfect agreement with experiment, qualitatively reproduces the key features and does not introduce unphysical instabilities. We conclude that $U = 0$ is more appropriate for describing the phonon behavior in the metallic phase of $K_2Cr_8O_{16}$.

6. The INS fits are not very convincing because there is very little contrast in the experimental data. Please, show the constant-energy cuts along with the spin-wave fits in order to confirm the accuracy of the proposed model.

Author Reply 6:

We thank the referee for this valuable comment. We agree that the contrast in the INS data is limited, which is primarily due to powder averaging effects. Additionally, some spurious signals in the data further complicate the comparison between experiment and model. However, despite these challenges, we believe there is sufficient contrast in the measured spectra to demonstrate the validity of the fitted spin-wave model.

To address the referee's suggestion, we have added constant-energy and Q cuts along with the corresponding spin-wave fits in the revised SM (Fig. S6). These additional figures more clearly illustrate the agreement between the experimental data and our model, and support the reliability of the extracted exchange parameters (which are comparable to the calculated values).

7. Calculated magnetic interactions are proportional to t^2/U , but the reasons for it are not explained in detail. The t^2/U dependence is typical for antiferromagnetic superexchange, but J1-J3 are all ferromagnetic. Which hopping parameters are considered in this case? I could imagine that FM superexchange follows the t^2/U dependence if 't' is the t_{2g-e_g} hopping, but is it really the case? What about the $t_{2g-t_{2g}}$ hoppings? They must lead to antiferromagnetic superexchange interactions.

Author Reply 7:

We thank the referee for raising this important point regarding the origin of ferromagnetic exchange and its relation to the t^2/U dependence. While it is true that the t^2/U scaling is typically associated with antiferromagnetic superexchange (particularly for $t_{2g-t_{2g}}$ or e_g-e_g interactions), ferromagnetic exchange can also exhibit a similar energy dependence under certain conditions, as described by the Goodenough-Kanamori-Anderson (GKA) rules.

$K_2Cr_8O_{16}$ is a mixed-valent compound containing both Cr^{3+} (d^3) and Cr^{4+} (d^2) ions in octahedral coordination. The relevant electronic states arise predominantly from the t_{2g} manifold, as the e_g levels are significantly higher in energy and are unoccupied. The Cr-O-Cr bond angles are close to 97° , leading to nearly orthogonal orbital overlap between neighboring Cr ions and strongly suppressing direct $t_{2g-t_{2g}}$ hopping.

Under these conditions, the GKA rules predict that ferromagnetic superexchange can emerge due to Hund's coupling on the intermediate oxygen ligand—specifically, the oxygen p orbitals. This mechanism, sometimes referred to as Hund's-rule-driven ferromagnetic superexchange, favors parallel spin alignment when electrons hop from a half-filled orbital (Cr^{3+}) to an empty orbital (Cr^{4+}) via an oxygen p orbital, especially when the orbital overlap is weak or orthogonal. Although this exchange differs from conventional AFM superexchange, it remains a second-order virtual hopping process. As such, the resulting exchange interaction retains a quadratic dependence on the hopping amplitude, scaling as t^2/Δ , where Δ is an effective excitation energy that includes the oxygen site energy and Hund's exchange. Therefore, while the J_1 – J_3 interactions are ferromagnetic in sign, they still follow a t^2/U -like dependence, as they are governed by the same underlying second-order perturbative framework.

Revision:

We have now added expanded the explanations mainly to the supplementary materials (Sec. S3C), in which the main text references.

8. Ref. [19] is not displayed in the manuscript. I presume it is New J. Phys. (2020)

Author Reply 8:

We thank the referee for pointing out this mistake. Reference 19 is indeed New Journal of Physics (2020). We have corrected this mistake.

Report of the third Referee

$K_2Cr_8O_{16}$ ferromagnetic hollandite has been studied for almost two decades for presenting the prototype ferromagnetic metal-insulator transition, a rare phenomenon. In two seminal PRLs in 2009 and 2011 (cited as refs. 16, 17 in the manuscript), and also in another paper the year after (ref. 18), the following description was given:

$K_2Cr_8O_{16}$ was determined to be a quasi-1D system with Peierls distortion in the ferromagnetic insulating state. Peierls here is due to the dimerization of tetramers within the unit cell with no doubling of the lattice, but it does not bring along charge ordering on the Cr sites. Ab initio calculations describe a double exchange mechanism between double chains via corner-shared oxygens, with exchange constants of 5.3 meV within the double chains but 10.6 meV between the double chains. This results in a Fermi surface of strongly quasi-1D character with a nesting vector of (0,0,1) along the c direction. The low-temperature structure was described in the $P112_1/a$ space group. In general, there is a mixed Cr^{3+}/Cr^{4+} valence. This picture has become well accepted.

A decade later, two ab initio studies—one cited as ref. 22 from NJP in 2020, the other from the same group in PRL the year before, but for some reason not cited (the manuscript refers to ref. 19 for a similar study, but it is actually missing from the reference list, probably a typo) — already described Weyl fermions in the system.

The present manuscript describes a different picture for the MIT. It presents experimental arguments against a Peierls instability and argues that the Weyl fermions bring along a correlation-induced topological phase transition that results in the MIT. The low-temperature magnetic structure is described in the $C2'/m'$ space group, surprisingly, since it doesn't seem to be the best choice for ferromagnetic ordering.

The manuscript also argues against the interaction being double exchange, based on calculations of the dependence of the exchange on the hopping integral, and instead argues for a uniform $Cr^{3.75+}$ valence state with superexchange-like interactions between them.

Yet, the manuscript proposes exchange values of 6 and 10 meV, quite similar to those from 2011.

Therefore, the current manuscript proposes a drastically different chemical situation from the original picture and a dramatically different origin for the MIT.

Author Reply:

We thank the referee for the accurate historical overview, as well as pointing out the reference typo. Reference 19 is indeed New Journal of Physics (2020). We have corrected the syntax error. In addition, we have included citation of the 2019 work from the same group in the Main Text:

“Related compounds such as $RbCr_4O_8$, have also shown to host Weyl points [Physical Review Letters **122**, 057205 (2019)], demonstrating the robustness of these topological features in this family of compounds. “

We agree that our picture presents different situation than what has been outlined in previous studies. We have addressed the questions raised, point by point, below.

1. A particularly glaring omission in the manuscript is that it strongly argues for the much improved quality of the samples, based on a modified high-pressure synthesis, but does not mention a single word about the modification. The original PRL from 2009 describes the high-pressure synthesis conditions in detail.

Author Reply 1:

We thank the referee for pointing this out. We have updated the manuscript to include the explanations about the modifications. New designs of 18/11 and 25/15 multi-anvil high-pressure cells were developed at Max Planck Institute, which provide stable experimental conditions: under 8GPa at 1500°C in 25/15 cell and 12 GPa at 1500°C in the 18/11 cell, allowing for larger sample volumes. We have included the following sentences to the manuscript:

“In particular, new cell designs for a high-pressure Walker-type multi-anvil system were developed at the Max Planck Institute for Solid State Research, enabling growth of larger, higher-quality crystals. The new 18/11 and 25/15 multi-anvil cell assemblies provide more uniform pressure and temperature distributions, resulting in increased sample volumes while maintaining stable synthesis conditions up to ~12 GPa and 1500 °C. With these improved cells the CrO₂ impurity phase is reduced to below 2 % in powder, and the obtained crystals are significantly larger (up to 0.5×0.5×1 mm³). The synthesis conditions otherwise follow those reported previously [Physical Review Letters **103**, 146403 (2009)].”

2. The manuscript proposes to set up a strong and interesting polemic in the literature. The pros and cons should be more clearly and antagonistically described; the arguments in the current text are quite scattered. However, and more importantly, there are both experimental and *ab initio* arguments in the current study. The *ab initio* results don't seem to be qualitatively different from the 2019-20 results.

Author Reply 2:

We thank the referee for this thoughtful comment and for pointing out the need to more clearly articulate the contrast with earlier work. We agree that our manuscript sets up an important discussion in the literature, and we have revised the text to highlight this polemic more explicitly and in a more structured way.

While our *ab initio* calculations share some qualitative similarities with the 2019-20 results [Physical Review Letters **122**, 057205 (2019); New Journal of Physics **22**, 073062 (2020)], our work differs in several key respects:

1. **Experimental anchor.** In our study, the experiments take the lead role, while first-principles calculations play a supporting role. With the aid of neutron scattering results, we carried out a group-theory analysis of the magnetically ordered states. This experimental anchor allows us to move beyond purely computational predictions and to

ground our symmetry analysis in experimentally verifiable measurements. For example, without experimental guidance, we would not have known that the magnetic space group of the metallic phase is #12.62, which laid the basis for our theoretical group-theory analysis.

2. **Peierls transition.** A central pillar of our work is the experimental disproof of the Peierls mechanism, which has been widely assumed in the literature. Earlier theoretical work did not test this assumption [Physical Review Letters **109**, 076401 (2012); New Journal of Physics **22**, 073062 (2020)], which constrained their assumptions and, consequently, their conclusions.
3. **Exchange interactions.** It was widely accepted that double-exchange interaction is the FM exchange interaction in the insulating phase [Physical Review Letters **109**, 076401 (2012)], based on the assumed Peierls transition. With Peierls transition disproved by our IXS measurements, the double-exchange scenario can also be questioned. We compared the exchange parameters J of nearest-, next-nearest-, and next-next-nearest neighbors with experiment, finding good agreement. By computing J vs. t/U , we find a quadratic dependence, allowing us to conclude that superexchange interactions govern the exchanges in both the metallic and insulating phases.
4. **Relation to topology.** The New Journal of Physics **22**, 073062 (2020) study established Weyl fermions in the metallic phase but did not consider the effect of the MIT. By contrast, we note that in the monoclinic phase, the combination of structural distortion and orbital splitting removes the Weyl points present in the ferromagnetic metallic phase. Our combined experimental–theoretical analysis thus demonstrates that the MIT links a Weyl semimetallic state to a trivial insulating state, establishing it as a correlation-driven topological MIT.

In the revised manuscript, we have reorganized parts of the discussion to make these contrasts clearer.

3. The experimental results of the inelastic X-ray scattering show a lack of phonon condensation. This is employed as an argument against the Peierls instability. However, the Peierls distortion proposed in 2011 is quite unusual. Would it necessarily be accompanied by drastic changes in the phonon spectrum?

Author Reply 3:

We thank the referee for this thoughtful question. As the referee notes, the Peierls instability—originally described by Peierls in the 1930s—refers to a transition in which 1D metallic chains become unstable toward lattice distortions due to electron-phonon coupling. This distortion leads to a gap opening at the Fermi surface and a transition to an insulating state. Crucially, such a transition is inherently accompanied by changes in the phonon spectrum (as the electron-phonon coupling is the foundation behind the transition), typically manifested as phonon softening or condensation at a characteristic wavevector.

Over time, more complex charge density wave (CDW) transitions have been observed, including those in higher dimensions (2D and 3D), where electronic susceptibility does not diverge and only partial gap openings occur. In these cases, the resulting ground state is not necessarily insulating. The term "Peierls transition" is typically reserved for the 1D case, where the distortion is tightly coupled to the divergence in electronic susceptibility and phonon condensation plays a central role, resulting in the insulating state.

Even in more modern theories of CDW formation—such as those involving momentum-dependent electron-phonon coupling—phonon anomalies are still expected, as the underlying mechanism remains electron-phonon coupling. Therefore, we argue that the lack of any clear phonon softening or anomaly in our inelastic X-ray scattering (IXS) data provides strong evidence against a Peierls-like transition in $\text{K}_2\text{Cr}_8\text{O}_{16}$.

We acknowledge, however, that a scenario involving the softening of an optical phonon might not result in full phonon condensation. Still, one would expect some anomaly in the phonon spectrum. However, previous theoretical work [Physical Review B **90**, 045124 (2014)] suggests that phonon condensation in $\text{K}_2\text{Cr}_8\text{O}_{16}$ is predicted for sufficiently high values of U , where an acoustic phonon softens at the $\mathbf{q}_{\text{CDW}} = (0.5, 0.5, 0)$ point. However, as also noted by Referee 2 in their Comment 5, our experimental data do not show any such anomaly.

We would also like to comment on the original proposal of a Peierls transition in $\text{K}_2\text{Cr}_8\text{O}_{16}$ [Physical Review Letters **107**, 266402 (2011)], which we believe contains a conceptual inconsistency. The nesting vector observed in that study is along the c -axis ($\mathbf{q}_{\text{nest}} = (0, 0, 1)$), a result we also confirm in our calculations. However, the same study observed lattice distortion propagates in the ab -plane, with $\mathbf{q}_{\text{CDW}} = (0.5, 0.5, 0)$. This means that the nesting and distortion vectors are perpendicular to one another. Thus, the observed nesting does not drive the lattice modulation. To argue for a Peierls mechanism, one would need to identify nesting along \mathbf{q}_{CDW} , but the Fermi surface in that direction is strongly 3D-like (see e.g. in Physical Review B **80**, 024416 (2009)). As discussed earlier, CDWs in three dimensions do not typically drive a full insulating state and are, by definition, not Peierls transitions.

We did not include these critiques and mistakes of earlier work in the main text, as our focus was to emphasize our own findings rather than highlight the shortcomings of previous studies.

4. The inelastic neutron scattering shows a lack of drastic change in the spin-wave excitations across the MIT. It is unclear—although the text makes it look as such—how this relates to the lack of a Peierls instability.

Author Reply 4:

We thank the referee for raising this important point. We agree that the inelastic neutron scattering (INS) results, on their own, do not directly demonstrate the absence of a Peierls instability. Our intention was to emphasize that the nature of the underlying structural units—being chimney-like rather than quasi-one-dimensional—provides an initial indication against the presence of a Peierls mechanism. Since a Peierls transition typically requires a quasi-1D electronic

structure, the dimensionality inferred from the INS data serves as a crucial contextual element for interpreting the IXS results.

We have revised the manuscript to clarify this point and to ensure that the connection between the INS findings and the broader discussion of the Peierls instability is not overstated.

To summarize, our argument against a Peierls instability in $\text{K}_2\text{Cr}_8\text{O}_{16}$ is supported by three key observations:

1. The crystal structure consists of chimney-like units rather than a 1D or quasi-1D lattice.
2. The previously proposed nesting vector ($\mathbf{q}_{\text{nest}} = (0, 0, 1)$) is perpendicular to the observed CDW vector ($\mathbf{q}_{\text{CDW}} = (0.5, 0.5, 0)$).
3. There is no clear Fermi surface nesting along $(0.5, 0.5, 0)$
4. There is no evidence of phonon softening or condensation associated with the transition.

These observations, taken together, strongly support the conclusion that a Peierls mechanism is not operative in $\text{K}_2\text{Cr}_8\text{O}_{16}$.

Reviewer #1 (Remarks to the Author):

I am very pleased to report that all of my concerns were indeed addressed by the authors of this manuscript. My primary concern was that wording including "paramagnetic" was included frequently and I had thought it was being used to describe calculations that were non-magnetic. As it turns out, there are various crystallographic and magnetic phases and two of them are monoclinic and experimentally there are paramagnetic phases, but all calculations were magnetic. Likely, having performed all of this work, the authors understood what they were referring to and where throughout the manuscript, but from an outsider's point of view, it was quite confusing. The authors have taken the time to extensively explain what has been done in their response to my report and then, crucially, have included relevant wording in the manuscript and SM.

I can now endorse publication of this work as it is indeed interesting and will surely have an impact on the community.

Reply:

We are grateful to the referee for the thoughtful comments and for giving us the opportunity to clarify and strengthen our manuscript. We appreciate the referee's positive evaluation and support of our work for publication in *Nature Communications*.

Reviewer #2 (Remarks to the Author):

Most of my previous concerns have been addressed in the revised version. However, I have some additional queries that are inspired by the clarified computational methodology:

Reply:

We thank the reviewer for the careful re-evaluation of our revised manuscript and for the additional comments inspired by the clarified computational methodology. We appreciate the opportunity to further improve our work, and we have addressed each point below.

1. Both abstract and introduction put a lot of weight onto the correlated electron physics, but I am tempted to ask what the manifestations of the correlated physics are. In fact, the authors argue that the metallic phase of $K_2Cr_8O_{16}$ is best described using $U=0$, so they essentially conclude on its uncorrelated nature. One especially odd statement is the claim of the "correlation-induced topological phase transition" in the abstract, while the rest of the manuscript never mentions correlations at all. In my opinion, the claim of an MIT driven by topology instead of correlations would be a much better framework for this study, at least in its current form (but see #2 below). I would also advise the authors to completely re-write the abstract and highlight the concrete experimental findings, such as the absence of phonon softening, instead of the vague and arbitrary concepts like "quantum devices", which do not occur in this study at all.

Reply 1:

We thank the reviewer for raising the question regarding the interplay between topology and electronic correlations. We have revised the abstract to better reflect the role of electronic correlations. Following the reviewer's suggestion, we have also updated the Supplementary Materials to explicitly state that we do not rule out the possibility of small but finite values of U (see Reply 2).

Importantly, upon cooling, a ferromagnetic insulating phase emerges. This crystal phase requires a finite on-site Hubbard correction in the DFT electronic band structure ($U > 3$ eV) in order to open the gap. The necessity of a nonzero U clearly indicates that local electronic correlations beyond standard DFT are essential for stabilizing the insulating state. The metal-insulator transition therefore has a correlation-driven aspect to it, and is accompanied by a change in the topological character of the bands. In this context, it is appropriate to describe the transition as being driven by electronic correlations and accompanied by a change in topology. Since the rest of the paper is based on this premise, electronic correlations were not discussed further unless necessary.

2. On a related note, the authors argue that the INS spectra do not change across the MIT, so the exchange couplings must be similar in the metallic and insulating phases. This is indeed the case in Fig. S9b,c if the $U=0$ results are compared, but it would then indicate that the insulating phase must be uncorrelated too. On a closer look, the spin-wave fits are done with $J_1=6$ meV and $J_3=10$ meV, the values that seemingly

correspond to $U=2$ eV in Fig. S9b. In contrast, the exchange couplings of up to 40 meV, as shown in Fig. S9a, should be inconsistent with the experiment. All of this is quite confusing and in fact casts doubt on the idea that the metallic phase is uncorrelated. Would not it be better to apply the same U -value to the metallic and insulating phases, considering that their exchange couplings are similar?

Reply 2:

We thank the reviewer for raising this important point. As correctly noted, while our calculated exchange constants reproduce the experimental trend ($J_3 > J_1 \gg J_2$), their absolute magnitudes differ from the experiment. As the reviewer may know, $U = 0$ eV does not imply that the system is completely uncorrelated; it indicates that the exchange-correlation functional in standard DFT already provides an adequate description of the metallic phase, for which itinerant carriers strongly screen the local Coulomb interaction. In contrast, the exchange-correlation of the insulating phase cannot be captured within standard DFT and therefore requires an explicit on-site Hubbard correction, marking a qualitative change in the nature of electronic correlations across the MIT.

With that said, we can note that the electronic structure and qualitative conclusions of our study do remain unchanged even if a moderate value of U (e.g., $U = 2$ eV) is included in the metallic phase. However, as reported in *Phys. Rev. B* **90**, 045124 (2014), introducing $U = 3$ eV leads to a phonon condensation that contradicts our experimental observation of the absence of phonon softening. Therefore, these results suggest that U in the metallic phase should remain at least below 3 eV. We have therefore revised the Supplementary Materials (Sec. S3C) to make this point explicit by adding the following clarification:

“In contrast, for the metallic tetragonal phase, no Hubbard correction was applied (i.e., $U = 0$ eV). While calculations with $U = 0$ eV reproduce the experimental trends, we cannot exclude the possibility that on-site corrections that include dynamical or multi-configurational effects may be needed to reproduce experimental magnitudes. Such effects would require approaches beyond DFT+ U , such as dynamical mean-field theory (DMFT), which goes beyond the scope of this work.”

3. A comparison of the band structures calculated with different ' U ' is shown in Fig. 3 of the response letter, but it does not appear anywhere in the manuscript. In my opinion, it should be included. I am also confused by the panel 'e' of that figure. What does it show exactly? What is the color scheme?

Reply 3:

We thank the reviewer for pointing this out. We have now included the comparison of the electronic band structures calculated with different on-site interaction strengths U in the revised Supplementary Materials (new Fig. S9). It shows that other than the slight modifications of the dispersion, the qualitative features of electronic band structure remain largely the same.

We apologize for the confusion over panel 'e'. The cross-like feature of Fig. R3e is equivalent to Fig. S8, but calculated with Hubbard U corrections. It uses the same colorbar as Fig. S8, where color intensity scales as $[\log(\text{gap})]^2$.

4. I would still recommend updating the captions of Figs. 3 and S7 and writing explicitly the U-values used in each calculation. This will facilitate the reading.

Reply 4:

We thank the referee for this recommendation. We have updated the figure captions, explicitly listing the U values used in each calculation.

5. It feels a bit strange that the magnetic couplings are in the range of 10 meV, but the INS data are mostly shown up to 4 meV only. The data obtained with the incident energy of 97 meV are also presented in Fig. S4, but without any comments or analysis, although these data clearly show more features than the low-energy data (note, for example, the flat band around 40 meV). Are the high-energy INS data equally well described by the spin model?

Reply 5:

We thank the referee for raising this question. We agree with this comment, and we have included calculation results ($J_1 = 6$, $J_3 = 10$ and $J_2 = 0.6$ and 0.4 meV) for the higher energies as well, in Fig. S4(g, h). As seen, the high intensity region around 60 meV, the flat features around 55 meV, the low energy low q intensity as well as the dispersion merging from 2.6 \AA^{-1} are reproduced by the model. We have appended the relevant part of the figure here below (Fig. R1). We have updated the SM by adding the paragraph:

Revision:

The exchange parameters obtained from the fits are summarized in the main text. The calculated spectra based on these parameters reproduce the experimental data well, capturing both the flat-band-like feature around 55 meV and the emerging dispersion near 2.6 \AA^{-1} (Fig. S4(g,h)). This confirms that the dominant magnetic units are groups of four corner-sharing chains ("chimneys"), as described in the main text. The consistent match between experiment and theory over the full energy range demonstrates that the extracted parameters are robust and physically meaningful.

Figure R1. (g, h) Calculated inelastic neutron scattering spectra obtained from the best-fit Heisenberg Hamiltonian for 5~K and 130~K, as defined in the main text.

6. Fig. S5 is hard to read because of the many black lines that look more like "spaghetti's". One could use different shades of gray for each line, and the values of J_2 should be given either in the graph or in the figure caption.

Reply 6:

We thank the referee for mentioning this point. We agree and have updated the figure, which is also appended below (Fig. R2). We also explicitly mentioned the J_2 values in the figure caption.

Figure R2. **A cut of the inelastic neutron scattering data.** A $Q = 0.6(1) \text{ \AA}^{-1}$ cut of the data shown in Fig. S4(a,b) for $T = 5 \text{ K}$ (blue) and $T = 130 \text{ K}$ (red), collected at $E_i = 9 \text{ meV}$. The solid and shaded areas show the calculated dispersion relations for $J_1 = 6 \text{ meV}$ and $J_3 = 10 \text{ meV}$ as a function of J_2 (from 0 to 1 meV in steps of 0.1 meV). The increase in intensity at lower energies originates from quasi-elastic scattering. The excitation shows a strong dependence on J_2 , with higher values shifting the excitation to higher energies.